# An aerosol climatology for global models based on the tropospheric aerosol scheme in the Integrated Forecasting System of ECMWF.

Alessio Bozzo[1*], Angela Benedetti[1], Johannes Flemming[1], Zak Kipling[1], and Samuel Rémy[1,2]

[1]ECMWF, Reading, UK
[2]IPSL, UPMC-CNRS, Paris, FR
[*]now at EUMETSAT, Darmstadt, DE

*Correspondence to:* Alessio Bozzo (alessio.bozzo@eumetsat.int)

**Abstract.** An aerosol climatology to represent aerosols in the radiation schemes of global atmospheric models was recently developed. We derived the climatology from a reanalysis of atmospheric composition produced by the Copernicus Atmosphere Monitoring Service (CAMS). As an example of application in a global atmospheric model, we discuss the technical aspects of the implementation in the Integrated Forecasting System of European Centre for Medium Range Weather Forecasts (ECMWF-IFS) and the impact of the new climatology on the medium-range weather forecasts and one-year simulations. The new aerosol climatology was derived by combining a set of model simulations with constrained meteorological conditions and an atmospheric composition reanalysis for the period 2003-2013 produced by the IFS. The aerosol fields of the reanalysis are constrained by assimilating the aerosol optical thickness (AOT) retrievals product by the MODIS instruments. In a further step, we used modelled aerosol fields to correct the aerosol speciation and the vertical profiles of the aerosol reanalysis fields. The new climatology provides the monthly-mean mass mixing ratio of five aerosol species constrained by assimilated MODIS AOT. Using the new climatology in the ECMWF-IFS leads to changes in direct aerosol radiative effect compared to the climatology previously implemented, which have a small, but not- impact on the forecast skill of large-scale weather patterns in the medium-range. However, details of the regional distribution of aerosol radiative forcing can have a large local impact. This is the case for the area of the Arabian Peninsula and the northern Indian Ocean. Here changes in the radiative forcing of the mineral dust significantly improve the Summer monsoon circulation.

## 1 Introduction

Aerosols have an important impact on the radiative budget of the Earth-Atmosphere system. They participate in the atmospheric radiative transfer directly by scattering and absorbing electromagnetic radiation and indirectly by interacting with cloud microphysics (e.g. Haywood and Boucher, 2000; Bellouin et al., 2005). The uncertainty in the total radiative forcing by natural and anthropogenic aerosols remains large (Boucher et al., 2013) and most recent global climate models include more or less sophisticated prognostic aerosol schemes to explicitly take into account the direct radiative impact of aerosols on radiation and their interaction with cloud microphysics and other components of the Earth system (e.g. Bellouin et al., 2011; Donner et al., 2011; Stier et al., 2005). The impact of aerosols on the skill of numerical weather prediction (NWP) models is less clear (Baklanov et al., 2018; Mulcahy et al., 2014) and conclusions vary depending on the diagnostics used (Reale et al., 2011) and

on the spatio-temporal scales analysed (e.g. Rémy et al., 2015). Global and regional NWP models employ often an approximate treatment of aerosol radiative forcing based on a climatological description of their spatial distribution. This choice is mainly due to the fact that coupling an NWP to an atmospheric composition model with a significant number (usually O(10)) of additional prognostic variables increases significantly the computational burden of the system but it might not translate directly into a clear improvement of the forecast skill (Morcrette et al., 2011; Mulcahy et al., 2014). Moreover, extra difficulties arise when assimilating real time observations to constrain the initialization of the prognostic aerosol field because some species require an accurate prediction of their sources, as in the case of anthropogenic and natural fires. A realistic representation of the mean climatological distribution of the most important aerosols can already improve the forecast skill both on a regional scale and globally (Rodwell and Jung, 2008).

With an increasing availability of large computer resources and the improvement of chemical transport models, an increasing number of studies explored the impact of including various levels of complexity in the representation of aerosol radiative effect in NWP models (Baklanov et al., 2014). Mulcahy et al. (2014) concluded that including both direct and first indirect radiative effects of prognostic aerosols in a global NWP model results mainly in a reduction in radiation and temperature biases on regional scale, with limited impact on weather forecast skill. The representation of the aerosol-clouds interaction remains uncertain and so its impact on NWP models.

The largest impact on weather forecast skill of a prognostic aerosols scheme coupled to an NWP model is in case of events associated with large aerosol optical depths such as dust storms or wildfires. In these situations a realistic representation of the aerosol distribution differs significantly from the average climatology and it can improve forecasts locally, especially close to the surface. Additionally, feedbacks linked to the direct aerosol radiative forcing can affect the production of the aerosol itself (Rémy et al., 2015). Similarly, Toll et al. (2015) and Zhang et al. (2016) showed that capturing the distribution of aerosols during extreme fires events has a significant impact on near-surface weather forecasts for the affected areas.

In the operational configuration of the European Centre for Medium Range Weather Forecasts - Integrated Forecasting System (ECMWF-IFS) the aerosol direct radiative effect has always been treated using climatological aerosol distributions with no attempts at representing the interaction between aerosols and cloud microphysics (Table 1). The IFS has employed since 2003 a monthly-mean climatology of five main aerosol species based on one of the first multi-aerosol model simulations by Tegen et al. (1997) ("TG97" in the following) and this substituted an earlier simpler annual mean distribution based on Tanré et al. (1984) (Table 1). When the more detailed TG97 climatology was introduced, it improved the model forecast skills mainly on a regional scale but, thanks to tele-connection feedbacks, it also affected the large scale mean flow (Rodwell and Jung, 2008). The tropical regions and in particular the monsoon areas of Western Africa and India showed the largest sensitivity to the change in aerosol radiative forcing, resulting in improvements in the precipitation bias (Tompkins et al., 2005).

Prognostic aerosols were introduced in the IFS for the first time with the GEMS project in 2005 (Hollingsworth et al., 2008) as part of the development of a real-time operational assimilation and forecast capability for aerosols, greenhouse and reactive gases. The aerosol assimilation and forecast model (Morcrette et al., 2009; Benedetti et al., 2009) has been further refined in the subsequent MACC projects (Simmons, 2010) and it is now maintained and developed within the Copernicus Atmosphere Monitoring Service (CAMS) as a suite of on-line integrated modules for aerosol and chemistry in the IFS (Flemming et al.,

2015; Morcrette et al., 2009; Rémy et al., 2019). Morcrette et al. (2011) used an early version of the aerosol scheme in the IFS to explore the impact of coupled prognostic aerosol on the quality of the operational IFS forecasts. Both direct and indirect radiative effects were included, the latter impacting the number concentration of liquid cloud droplets according to Menon et al. (2002). They found that compared to the TG97 climatology, the changes in medium-range large-scale forecast skill caused by having the prognostic aerosols interacting with radiation and cloud microphysics were small, although near-surface parameters showed local improvements. The inclusion of the full prognostic aerosol model had a prohibitive impact on the efficiency of the IFS, increasing the whole computational cost of the model by more than 50%. However, no attempt at optimizing the implementation was made.

**Table 1.** Evolution in the treatment of radiative effect of aerosols in the ECMWF IFS forecast model.

| years in use | aerosol model | characteristics |
|---|---|---|
| 2000 - 2003 | Tanre et al. 1984 | 4 aerosol types (desert, continental, maritime, industrial), annual average, total integrated AOD |
| 2003 - 7/2017 | Tegen et al. 1997 | 5 aerosol types (dust, organic, sulfate, black carbon, maritime), monthly average, total integrated AOD |
| 7/2017 - present | CAMS | 5 main aerosol types (dust [3 size bins], organic, sulfate, black carbon, sea salt [3 size bins]). Monthly averages. Distinction between hydrophilic and hydrophobic species. Total integrated AOD before 2019 then mass mixing ratio profile at each grid point. |

A climatological description of aerosol distribution is still a viable option to capture the monthly-mean aerosol radiative effect for a NWP model (Toll et al., 2016). Improvements in aerosol climatologies are tied to improvement in chemical transport models and observations and it can be represented as a two- or three-dimensional spatial distribution of aerosol mass or optical properties. A climatology can be built with a strong emphasis on surface observations using model fields to fill the gaps between the sparse network of measurement sites (e.g. Kinne et al., 2013), or merging model fields, satellite data and surface observations using empirical methods (e.g. Liu et al., 2005). A further option is to rely on a data assimilation system, which is the approach followed in this work.

The MACC reanalysis of reactive trace gases and aerosols (MACCRA, Inness et al., 2013) was the first multi-year atmospheric composition reanalysis effort developed with the MACC system taking advantage of the 4D variational assimilation system for atmospheric composition (Benedetti et al., 2009). Total aerosol optical thickness (AOT) was constrained by assimilating the AOT retrieved from the Moderate resolution Imaging Spectroradiometer (MODIS) observations. CAMS is currently updating MACCRA with a new high-resolution atmospheric composition reanalysis (the CAMS reanalysis, CAMSRA) and as an interim product between MACCRA and CAMSRA, a new dataset (CAMS Interim reanalysis, CAMSiRA, Flemming et al., 2017) was produced. CAMSiRA shows a good agreement with the latest surface AOT observations, combines the most recent advances in global aerosol modelling and satellite retrieval and so it represents an improvement with respect to the

current TG97 climatology as well as MACCRA. It also provides a better framework to evaluate the impact of coupling the IFS prognostic aerosol model to the operational forecast system.

This document describes the development of a three-dimensional monthly-mean climatology of five aerosol species based on CAMSiRA (section 2). A comparison of the climatological values against daily aerosol optical thickness observations will be discussed in section 3. As an example of application, in section 4 we describe its implementation in the IFS discussing the impact on the climatology on the mean model climate and on its forecast skills. The aerosol climatology is intended for public use and it will be available through the CAMS data service.

## 2   CAMS aerosol climatology

The aerosol model implemented in the CAMS system is based on the model developed at the Laboratoire d'Optique Atmosphérique (LOA) Laboratoire de Météorologie Dynamique (LMD) (Boucher et al., 2002; Reddy et al., 2005) with modifications by ECMWF during the GEMS and MACC projects. Details of the model can be found in Morcrette et al. (2009), Benedetti et al. (2009) and Rémy et al. (2019). Only a brief summary is given here.

Five types of tropospheric aerosols are considered: sea salt (SS), dust (DU), hydrophilic and hydrophobic organic matter (OM), black carbon (BC) and sulfate (SU) aerosols. Prognostic aerosols of natural origin, such as mineral dust and sea salt are described using three size bins each (the size bins range is defined by the radius of the aerosol particle in microns, 0.03,0.55,0.9,20.0 for dust and 0.03,0.5,5.0,20.0 for sea-salt) represented by three separate prognostic variables each. Hygroscopic effects are taken into account for sulfates, sea salt and organic matter. This means that the CAMS system computes a total of 11 prognostic variables. Emissions of dust depend on the surface wind (as measured at 10m), soil moisture, the surface albedo in the UV-visible range and the fraction of snow-free land covered by vegetation, with a correction to account for wind gusts (Morcrette et al., 2008a; Rémy et al., 2019). The surface albedo in this case selects the area that can emit dust and weights the strength of the emission itself (Rémy et al., 2019) Emissions for Sea-salt depend on a source function based on Monahan et al. (1986) and representative at 80% relative humidity. Sources for the other aerosol types which are linked to emissions from domestic, industrial, power generation, transport and shipping activities, are taken from MACCity annual- or monthly-mean climatologies (Granier et al., 2011). Emissions of OM, BC and $SO_2$ linked to fire emissions are obtained using the GFAS system based on MODIS satellite observations of fire radiative power, as described in Kaiser et al. (2011). The OM species include contribution from organic carbon from biofuel, fossil fuel and biomass burning with a small contribution of secondary organic aerosols from biogenic sources (based on terpene emissions). Sulfate aerosol are linked to $SO_2$ emissions currently in a simple way parametrized in terms of temperature and relative humidity to allow for the representation of the diurnal cycle. Further details about the parametrization of the conversion rate between sulfur di-oxide and sulfate aerosols can be found in Rémy et al. (2019).

MODIS AOT data at 550 nm are routinely assimilated in a 4D-Var framework extended to include aerosol total mixing ratio as extra control variable using a variational bias correction based on the operational set-up for the assimilation of radiances

following Dee and Uppala (2008). The reader interested in the details of its implementation in the IFS should refer to Benedetti et al. (2009) and Benedetti and Fisher (2007).

As discussed in Flemming et al. (2017), the total AOT in CAMSiRA shows a good agreement with surface-based AERONET (Aerosol RObotic NETwork, Holben et al. (1998, 2001), https://aeronet.gsfc.nasa.gov, last access: 31 October 2019) observa-

tions. However, problems have been identified with the way the data assimilation distributes the contribution of the various species to the total AOT, in particular introducing unrealistic high sulfate burden over the oceans. We therefore derived the climatological distribution of the 11 prognostic CAMS aerosol types using the Control Run (CR) set up alongside CAMSiRA and covering the period 2003-2013. This experiment uses the same meteorological data and emission as CAMSiRA but without data assimilation, hence leaving the aerosol species free to evolve. We then used the total AOT from CAMSiRA to constrain

this climatological AOT by scaling the monthly mean distribution of the individual species to reproduce the total AOT computed in the reanalysis. Therefore, each monthly-mean AOT for the single species $i$ at the grid-point $(x, y)$ is adjusted following the simple relation:

$$AOT_{i,clim}(x,y) = \frac{AOT_{RA}(x,y)}{AOT_{CR}(x,y)} * AOT_{i,CR}(x,y) \tag{1}$$

where $AOT_{RA}$ indicates the total AOT at 550 nm from the reanalysis, $AOT_{CR}$ the total AOT from CR. Each species is

therefore scaled according to its contribution to the total AOT in a particular grid point.

The scaling computed from the AOT also applies to the mass mixing ratio, because consistent extinction coefficients are used between CR and CAMSiRA. The climatological scaled AOT is used only as diagnostic data in this work, while the actual aerosol climatology is computed in terms of a gridded monthly spatial distribution of layer-integrated mass concentration $[kg/m^2]$ for each aerosol component over 60 vertical levels. We provide the mass concentration per layer because it will be

directly proportional to the AOT given a mass extinction coefficient in $[m^2/kg]$, since we believe this would be the primary use of such a climatology. A set of gridded monthly-mean pressure profiles is also provided to allow the conversion to mass mixing ratio $[kg/kg]$ and the interpolation to other vertical grids. The native horizontal grid of CAMSiRA is a reduced Gaussian grid with 80 grid points between the Equator and the Poles (N80), equivalent to a linear grid resolution of approximately $1.125° \times 1.125°$.

For the native CAMSiRA horizontal grid the full 3D distribution over 60 vertical levels has a size of ∼3 Gb. The grid can be coarsened according to the desired resolution and we found that an horizontal grid of 3x3 degrees with a total size of ∼500 Mb was appropriate for the implementation in a relatively high resolution global model such as the ECMWF IFS with a spatial resolution equivalent to ∼ 9 km.

## 2.1 Spatial and vertical distribution of mass mixing ratio

The spatial distribution of the integrated mass for each individual type and for their sum shows marked regional and seasonal variations (Fig. 1). The largest contribution to the total global aerosol mass comes from mineral dust due to the large emissions over land especially across the arid areas of Northern Africa and central-east Asia in the Northern-Hemisphere summer months.

Sea salt is the most widespread species and it represents the second largest contribution to the total global mass, with the highest concentrations found in the storm track areas of the Northern and Southern Hemisphere. The organic matter and black carbon associated to the emissions from various anthropogenic and natural processes displays a large seasonal variation and highly localized regional distribution. The sulfates are mostly distributed over the Northern Hemisphere but the largest concentration is found close to the sources of anthropogenic emissions.

Each aerosol species exhibits a characteristic vertical distribution with a distinct seasonal cycle, shown in Fig. 2 as zonal-average profiles. Sea salt is confined close to the surface and it is strongly linked to the strength of the mid- and high-latitude winds with a separate maximum around 15 degrees N associated with the Indian monsoon. Mineral dust is transported vertically over the major desert areas of North Africa and Australia during the respective summer seasons, with a significant amount of mass up to 600 hPa in the Northern Hemispheric Summer. The seasonal variation in the strength of the anthropogenic and natural organic emissions controls the amount and vertical extent of the organic and black carbon species with the main source located around the Equator and a June-July-August maximum in the Northern Hemisphere linked to the fire season in the high latitudes. sulfate emission, mostly from the industrialized areas peak during the Northern Hemisphere Summer with a maximum just below 700 hPa.

A small amount (< 1e-3 $g/m^2$ per layer) of aerosol mass is present in the upper tropospheric and lower stratospheric layers, mostly at high latitude. This process is likely to be overestimated in the CAMS model due in large part to the lack of any effective removal process for high-altitude aerosol (except for coarse dust and sea salt which are subject to sedimentation). This means that any excess of aerosol mass that the model places in the upper troposphere/lower stratosphere tend to have a long residence time and it affect the way the assimilation scheme redistributes vertically the total optical depth increments. This small amount of stratospheric aerosol is therefore not to be considered as representative of the contribution of stratospheric injection from large volcanic eruption. The AOT linked to stratospheric volcanic aerosols needs to be provided separately as the present climatology only represents the tropospheric aerosol species.

It is also possible to compute a characteristic climatological scale height for each species to describe the bulk of their vertical extent. This could be used to compute a simplified climatological vertical distribution for applications that do not need a detailed description of the full 3-dimensional fields. Appendix A briefly discusses the details of the derivation of such a parameter from the CR. The result is shown in Fig A3 and, as seen in the zonal-mean profiles, it highlights regions where the aerosol species are transported away from the near-surface sources to higher levels.

Until recently the IFS used an implementation of the climatology from Tegen et al. (1997) which relied on an analytical function of type $(p/p_0)^{(H/\xi)}$ to redistribute vertically the aerosol optical depth. The function depends on the atmospheric pressure $p$, the surface pressure $p_0$ and the ratio between the scale height of the standard atmosphere $H = 8.4$ km and a fixed global-mean scale height $\xi$ for each aerosol component. A comparison of the vertical distribution of the aerosol mass in the CAMS climatology using this approach to the real mean 3-dimensional distribution is shown in Fig. 3. Although in general the vertical profile is reproduced reasonably well, the simplified analytical approach does not capture the elevated maximum between 850 hPa and 700 hPa observed in the model data and stretches too high the upper boundary of the distribution. For

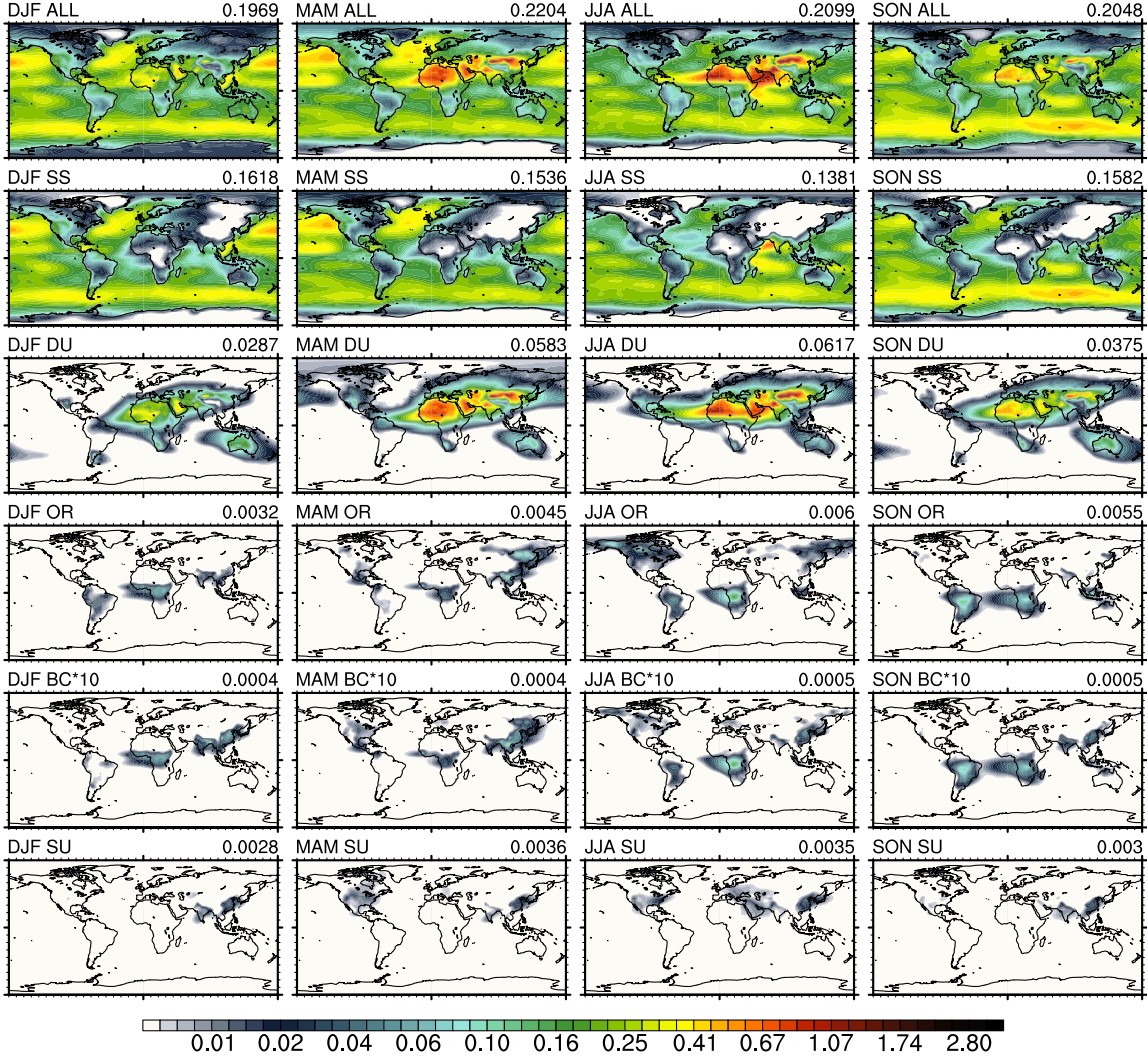

**Figure 1.** Seasonal vertically-integrated aerosol mass (g/m$^2$) from the CAMS Interim reanalysis control run, scaled to conserve the total AOT of the assimilation run. The top row shows the total integrated mass for all aerosol types and the other rows the contribution from the single species for each season (see text for the abbreviations). Indicated in the top right of each map is the global average. Notice that values for the black carbon type (BC) have been multiplied by 10 for better visualization while the global average shows the unscaled value.

species with not-negligible absorption in the solar spectrum such as mineral dust and organic matter, this means a vertical displacement in the solar heating rate profile which impacts the temperature profile.

## 2.2 Spatial distribution of optical thickness

When looking at the total aerosol optical thickness (AOT), the contribution from the various aerosol types depends on the combination of their mass load and their extinction efficiency. Since in principle any choice of optical properties can be associated to the climatological distribution of aerosol mass mixing ratio, we will not attempt here a thorough discussion of possible refractive indices and micro-physical models to describe the radiative properties of each aerosol species. Instead, we will briefly focus on the difference between the optical properties used for the CAMS aerosols in the implementation example discussed in the following sections and those used for the aerosol climatology employed until recently in the IFS.

Until cycle 43r3 (2017) the radiative effect of aerosols in the ECMWF IFS was computed using a monthly mean climatology of total AOT based on the total mass load from TG97. The AOT vertical profile was computed analytically with an exponential function as described in section 2.1 using a constant scale height $\xi$ for each species with $\xi = 3000$ for dust and $\xi = 2000$ for the other species. The climatology was coupled to the ECMWF radiation scheme (Morcrette et al. (2008a), Hogan and Bozzo (2018)) using optical properties derived from OPAC (Hess et al., 1998) and computed over a set of six coarse broad band intervals with no dependence on the relative humidity from the model and a spatial resolution of 4x5 degrees.

To implement the radiative effect of the CAMSiRA aerosols in the IFS we adopted the set of optical properties currently used in the IFS to diagnose the AOT from the CAMS aerosol forecasts. The details of the choices of size distributions and refractive indices for each species are discussed in more detail in Appendix B.

The largest differences in the optical properties used for the TG97 and CAMSiRA climatologies are found for the hydrophilic species organic matter and sulfates and for the hydrophobic mineral dust (Fig.A2). In the old climatology sulfates and organic matter aerosols were combined in a single species, resulting in a generally larger absorption in the short-wave range when compared to the separate contribution from organic matter and sulfates in the CAMSiRA climatology. The old optical properties for dust represented an average of what can be expected across the three size bins used in the CAMS climatology, but with significantly more absorption across the solar spectrum between 2.0 $\mu$m and 0.4 $\mu$m and a much smaller total extinction at infrared wavelengths.

The total AOT distribution for CAMSiRA climatology is shown in Fig. 4 compared to the AOT from the old TG97 climatology. The picture reflects the mass distribution seen in Fig. 1 with a strong seasonal variations both over land and over ocean. Compared to the AOT from the TG97 climatology, we notice a larger contribution from sea salt and a significantly different distribution over land. In particular the CAMS climatology has a larger AOT over the desert regions of Northern Africa and Central Asia as well as in the biomass burning areas of Central Africa, North America, Northern Asia and South America, the latter also showing a different seasonal cycle. The AOT is also larger over industrialized areas in India and Eastern Asia while it is smaller over Europe due to changes in the industrial emissions over these regions, dominated mostly by sulfates.

We observe even larger differences in the absorption AOT (AAOT, Fig. 5), resulting from the combination of changes in the mass distribution and in the optical properties. The CAMS climatology captures with a finer resolution the emission of

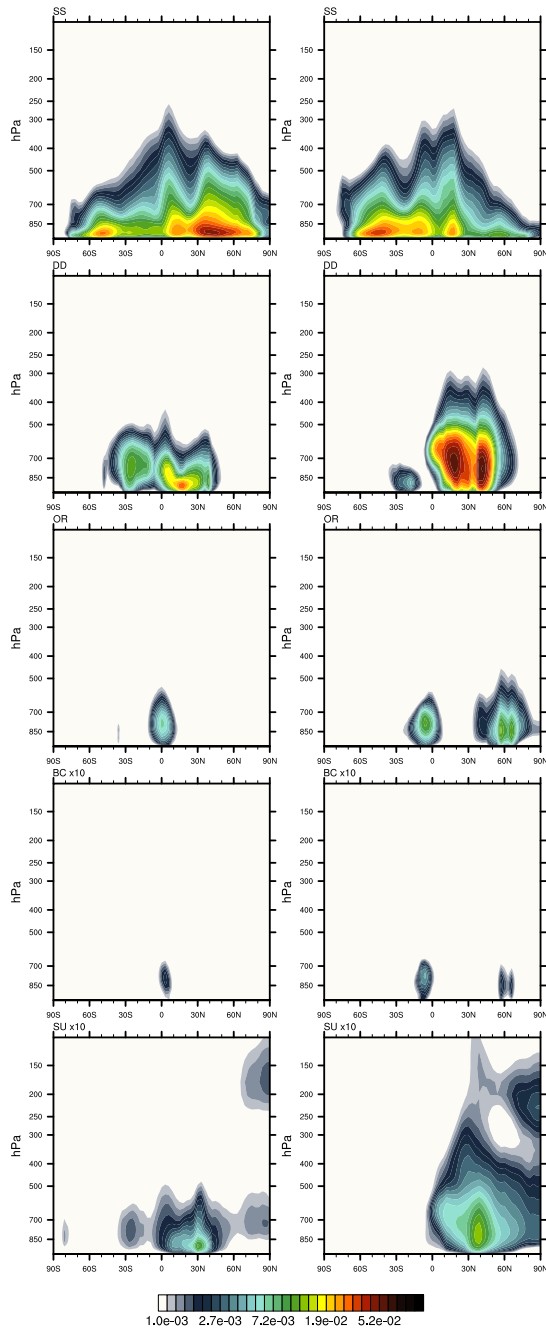

**Figure 2.** Zonal-mean layer-integrated mass profiles (g/m$^2$), weighted by the total integrated mass at every grid point. Monthly average for January (left) and July (right). Notice that for black carbon and sulfates values have been multiplied by 10 for better visualization.

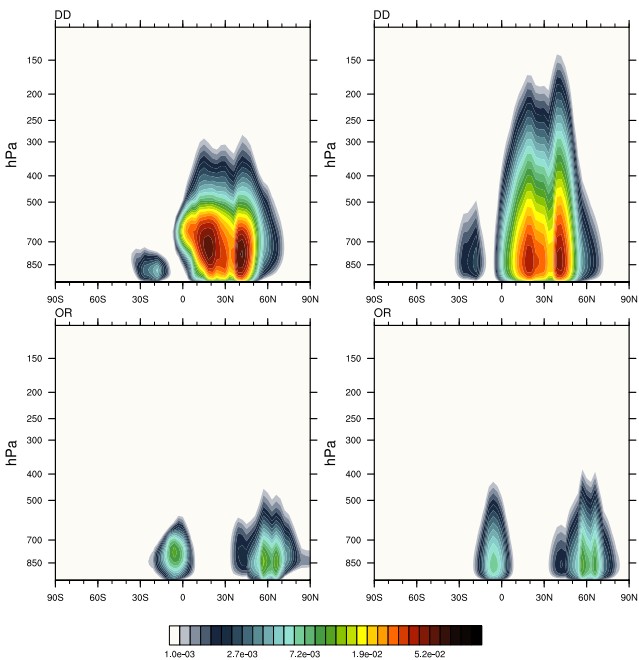

**Figure 3.** Zonal-mean layer-integrated mass profiles (g/m$^2$) average for July, weighted by the total integrated mass at every grid point for mineral dust (top row) and organic matter (bottom row). The left column shows the zonal mean from CR fields, the right column shows the vertical profiles computed using an exponential function applied to the total integrated mass (see text). The scale height used for dust is 3 km while for organic matter it is 2 km.

black carbon and organic aerosols in Central and Eastern Asia while showing significantly less absorption over Europe. Also significantly different is the AAOT distribution over Africa and the Middle-East where the TG97 climatology has a maximum in JJA over the Horn of Africa while the CAMS climatology has its maximum over Central Africa and Western Sahara.

## 3 Verification of the climatological aerosol distribution against surface observations

5 The accuracy of the CAMSiRA reanalysis and the CAMS aerosol model is discussed in Flemming et al. (2017); Inness et al. (2019); Rémy et al. (2019) in terms of global and regional bias and correlation against surface observations in terms of AOT and particulate matter. Since the AOT computed from the climatology of mass mixing ratio discussed in this work is based on the same model, it has the same mean bias as the CAMS Interim reanalysis when evaluated over a multi-year time interval. What we want to show here is how a monthly mean climatology compares to a full prognostic aerosol scheme in terms of daily 10 and intra-annual variability of aerosol distribution for a particular year.

To answer this question we used the observation from the AERONET network (Holben et al., 1998, 2001) for the year 2008 and compare them to the aerosol fields from the monthly mean CAMS climatology and from the AOT for 2008 in the CAMS

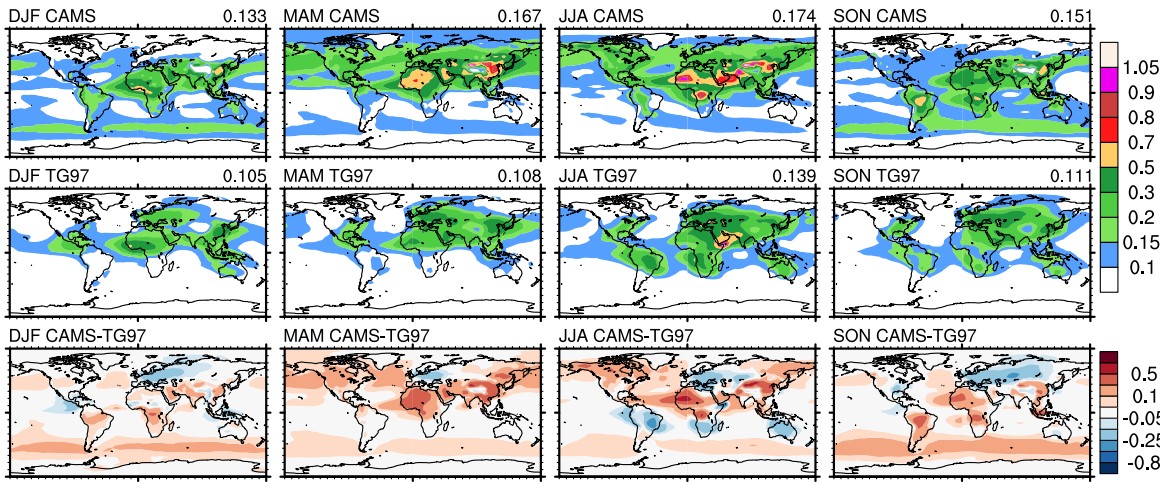

**Figure 4.** Seasonal total extinction aerosol optical thickness at 550nm. CAMS Interim reanalysis control run, scaled to conserve the assimilated AOT (top row) and the TG97 climatology (middle row). The bottom row shows the difference between the CAMS and the TG97 climatology.

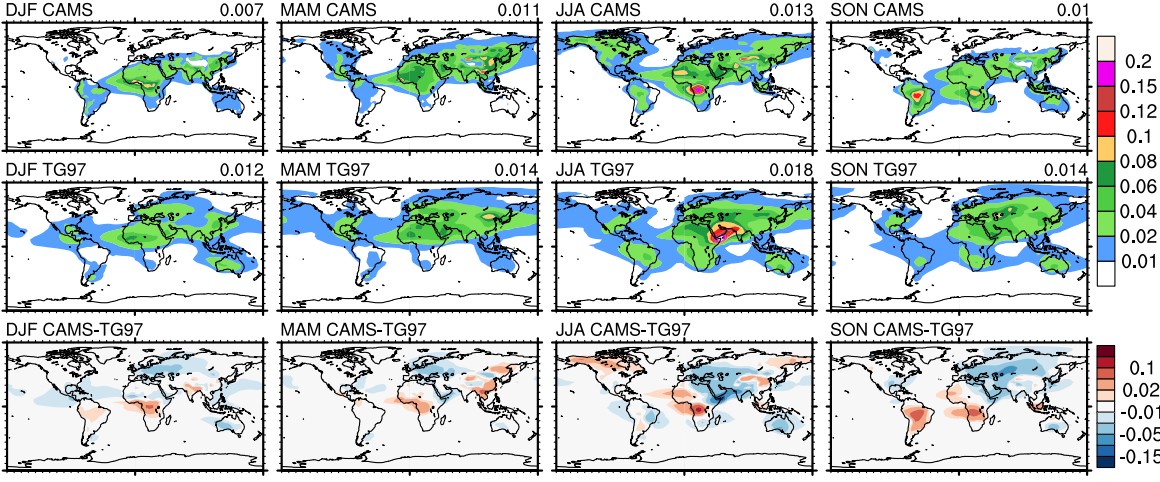

**Figure 5.** As Fig. 4 but for absorption optical thickness

Interim reanalysis. We also included in the comparison the TG97 climatology used in the IFS until CY43R3 (pre 2018). The monthly mean climatologies are linearly interpolated between the mid of each month.

The overall results over the whole globe and for five macro-areas are summarized in Table 2. The CAMSiRA climatology has a mean bias and correlation comparable to the corresponding reanalysis for the year 2008, with the lowest correlations observed over North America and Europe and the largest mean error over North and South America. Between the two climatologies CAMSiRA has in almost all of the regions higher correlations and lower mean bias than TG97.

The results from the global and regional scores can be understood looking more in detail at how a climatological description of the total AOT and the AAOT compares with the intra-annual variability observed in a particular year for a few locations (Fig. 7). We choose sites characterized by a dominant aerosol species and with different seasonal characteristics. The locations of the sites are shown in Fig. 6 together with the contribution of the five main CAMS aerosol species.

For the comparison in terms of AAOT we use the L2.0, version 2 of the almucantar retrieval at each AERONET site (Dubovik et al., 2002; Holben et al., 2006). Given that the retrieval producst are available with lower frequency than direct AOT observation, we used an average over the five years 2006-2010 to increase the data coverage. The evaluation of the AAOT strongly depends on the choice of the optical properties associated to the climatology and therefore in the case discussed here it mostly reflects the single scattering albedo applied in the implementation of the climatology within the ECMWF IFS (Appendix B). The absorption characteristics of the aerosol species are very sensitive to the refractive index to associate to each specie and a thorough discussion of the quality of different sets of optical properties is beyond the main scope of this work.

In the island of Midway in the Pacific Ocean, the AOT is dominated by sea salt with some contribution from sulfate aerosols and both climatologies have comparable total AOT. Since the optical properties implemented for the CAMS climatology depend on the local relative humidity, here and for the other sites as well the daily AOT computed from the CAMS monthly mean mixing ratio distribution inherits the synoptic variability of the humidity field. The comparison for the AAOT against AERONET is not shown because the aerosols dominating this site have a very low absorption at visible wavelengths, yet we can see that the implementation of the TG97 climatology has likely a too high mean absorption also in areas which should not be subjected to large amounts of absorbing species.

Chiang-Mai in northern Thailand represents a site with a seasonal influence from organic emission and industrial pollution in SE Asia. The seasonality in both AOT and AAOT is not captured in the TG97 climatology while the CAMS climatology, although clearly underestimates the peaks AOT for the year in question, does reproduce the period with large aerosol amount between February and May. On average the strength of the absorption is underestimated at the peak aerosol load.

In the Indian sub-continent both climatologies reproduce the seasonal variation on the AOT linked to the monsoon circulation and the anthropogenic emissions. Both climatologies tend to underestimate the total AOT in the northern India-Pakistan with the CAMS climatology having an overall lower bias. As an example, at the site of Karachi, influenced by dust and anthropogenic emissions, the two climatologies have comparable total AOT but the CAMS climatology shows a slightly larger AOT maximum between June and August. The AAOT curve shows much larger differences than for AOT between the two climatologies, with too much absorption for the TG97 climatology when compared against the AERONET retrieval for most of the year. Both climatologies seem to understimate the absorption during the winter month, despite a relatively low AOT bias.

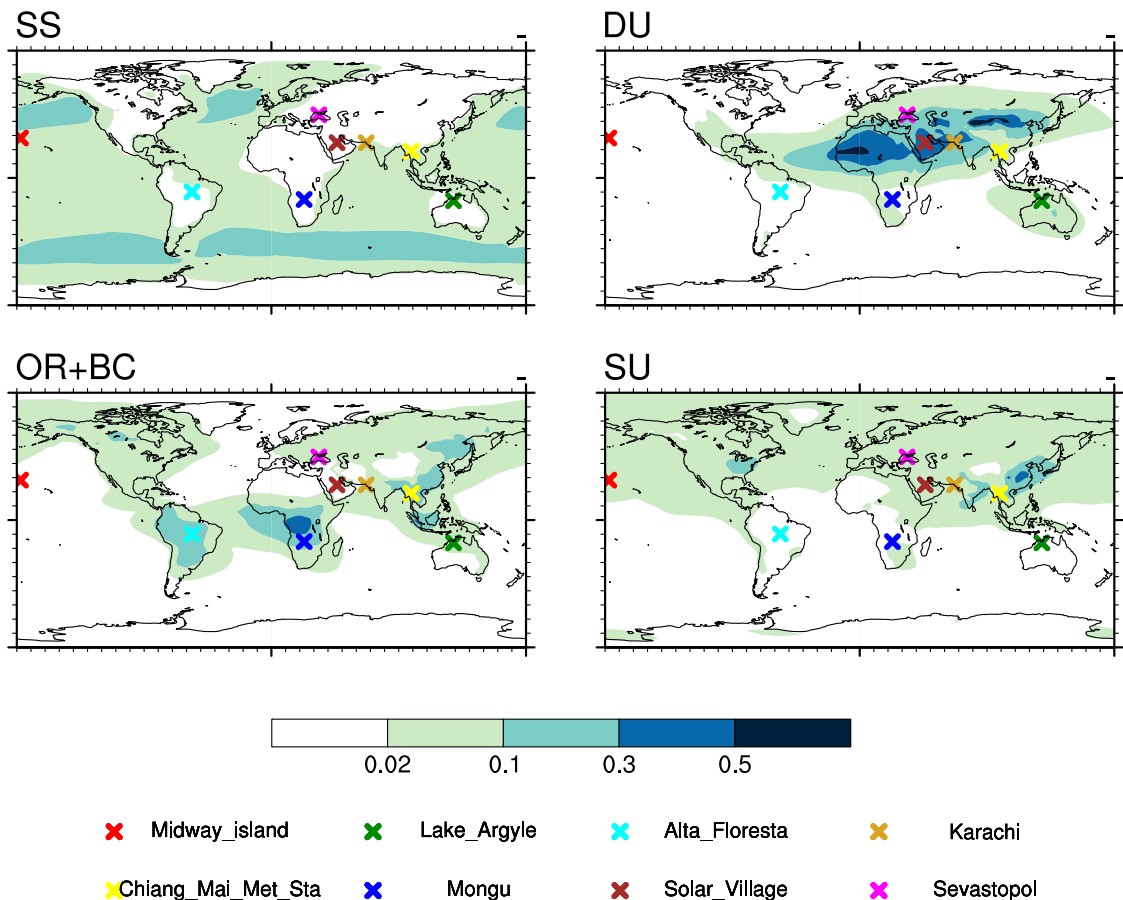

**Figure 6.** Location of the AERONET sites used in the analysis. The shading shows the annual mean AOT of the five CAMS aerosol species sea salt (SS), mineral dust (DU), organic matter (OR), black carbon (BC) and sulfates (SU).

Mongu, Zambia, shows a peak in AOT between August and October linked to the seasonal biomass burning. The AOT from CAMSiRA with fully prognostic aerosol does a good job in representing the daily variability linked to the organic species emissions while the climatologies are able to represent the increase in mean AOT from July-August to October while obviously missing the variability linked to the single events. The underestimation of the AOT results in a slightly underestimated AAOT near the August-October peak but it gives a good representation of the average absorption conditions at the site.

Other areas show a very different representation of the seasonal cycle in AOT between the two climatologies, especially over South America and Australia. For example at the Lake Argyle station, under the influence of mineral dust from the australian interior and organic matter to the north, the CAMS climatology does a good job in capturing the minimum AOT over the Southern Hemisphere winter months while TG97 appears out of phase, perhaps due to the now out-dated emission inventories used in those earlier aerosol transport experiments. Not many data are available for the AAOT comparison at this site, though the few close to the peak period September-November suggest an understimation in the total absorption by both climatologies.

Again in the southern Hemisphere the station of Alta Floresta in South America shows a strong seasonal peak during the biomass burning season August-November. Although the CAMS climatology does capture this peak fairly accurately, it nevertheless overestimates it both in term of total AOT and especially AAOT. The fact that AAOT has a large positive bias is partially due to the overestimation in the AOT and possibly partially due to too large absorption associated to the organic and black carbon species. This is related to some of the problems with the representation of biomass burning events in the CAMS model, as discussed in the section 4.3.

Dominated by desert dust, Solar Village in Saudi Arabia shows a very variable AOT timeseries reflecting the nature of large dust plumes measured in the region. The CAMS climatology has a larger AOT than TG97 with a seasonal cycle capturing the larger dust activity over the April-August period. The AAOT in the CAMS climatology slightly overestimates the average conditions retrieved by AERONET but does significantly better than the TG97 climatology which greatly overestimates the absorption between May and August. This is an important difference which can have a large impact on local atmospheric circulation, as discussed later in section 4.4.

A final example representative of Central and Eastern Europe where industrial aerosols dominate, the observations at Sevastopol on the Crimean peninsula reveal a systematic bias in the TG97 climatology with an overestimation in both the AOT and the AAOT for the whole year. This bias is likely linked to the industrial emissions used in TG97 for Europe based on the GEIA database relative to the year 1985 (Benkovitz et al., 1996) and not representative of the emissions in the period 2003-2013 togheter with a too high absorption characteristics associated to the industrial species.

## 4  The CAMS aerosol climatology in a global NWP model: implementation in the ECMWF IFS

As an example of the use of the CAMS aerosol climatology in a complex global atmospheric model, we briefly discuss its implementation in the ECMWF global weather forecasting system (IFS).

**Table 2.** Comparison between model and observation AOT for year 2008 at AERONET sites. Model values are from the prognostic AOT from the CAMS Interim reanalysis (CAMSiRA 2008) and two climatological fields from TG97 and from the current work (CAMSiRA CLIM). The columns report the mean error and the correlation coefficient for five macro areas and globally for all sites. The size of the sample used to compute these statistics is reported in brackets in the OBS column.

| | OBS 2008 | CAMSiRA 2008 | | Tegen et al. (1997) | | CAMSiRA CLIM | |
|---|---|---|---|---|---|---|---|
| | | mean err | corr | mean err | corr | mean err | corr |
| N America | 0.1341 (1425) | 0.032 | 0.6678 | 0.037 | 0.369 | 0.040 | 0.4907 |
| S America | 0.1603 (370) | -0.006 | 0.8356 | 0.047 | 0.316 | 0.040 | 0.814 |
| Europe | 0.176 (1112) | 0.009 | 0.6992 | 0.135 | 0.4187 | 0.015 | 0.4954 |
| Africa | 0.2689 (566) | 0.009 | 0.7926 | 0.004 | 0.5373 | 0.005 | 0.7395 |
| SE Asia | 0.4329 (780) | -0.056 | 0.6786 | -0.142 | 0.4165 | -0.07 | 0.6462 |
| Global | 0.215 (4259) | 0.004 | 0.77 | 0.025 | 0.439 | 0.009 | 0.725 |

We implemented in the IFS the full 3-dimensional CAMS climatology with the optical properties computed as described in Appendix B and we will discuss here the impacts of the new climatology compared to the IFS configuration using the old climatology based on TG97.

The implementation of the TG97 and CAMSiRA climatology in the IFS radiation scheme differ in a number of ways. Both
climatologies are represented in the radiation scheme in terms of the three bulk ratiative properties mass extinction coefficient $(m^2/kg)$, single scattering albedo and asymmetry parameter at each of the 30 spectral bands of the radiation scheme. For TG97 the optical properties were computed based on the OPAC database (Hess et al., 1998) at a fixed value of relative humidity.

For the CAMSiRA climatology the optical properties are computed according to the description in Appendix B. For the hydrophilic species the optical properties are interpolated to the relative humidity value provided by the IFS model at every
grid point and vertical level.

Moreover, as already mentioned, CAMSiRA climatology is defined for each grid point and vertical level while for TG97 an empirical vertical distribution is assumed (see section 2.1)

The different impact of the two climatologies in the IFS depends therefore on the combined effect of different aerosol spatial and vertical distribution and different radiative properties.

**4.1   Model experiments setup**

In the following sections we will show the impact of the CAMS climatology on the IFS both in terms of changes in mean bias and in forecast skill, as measured by the correlation between modelled and observed large-scale atmospheric circulation.

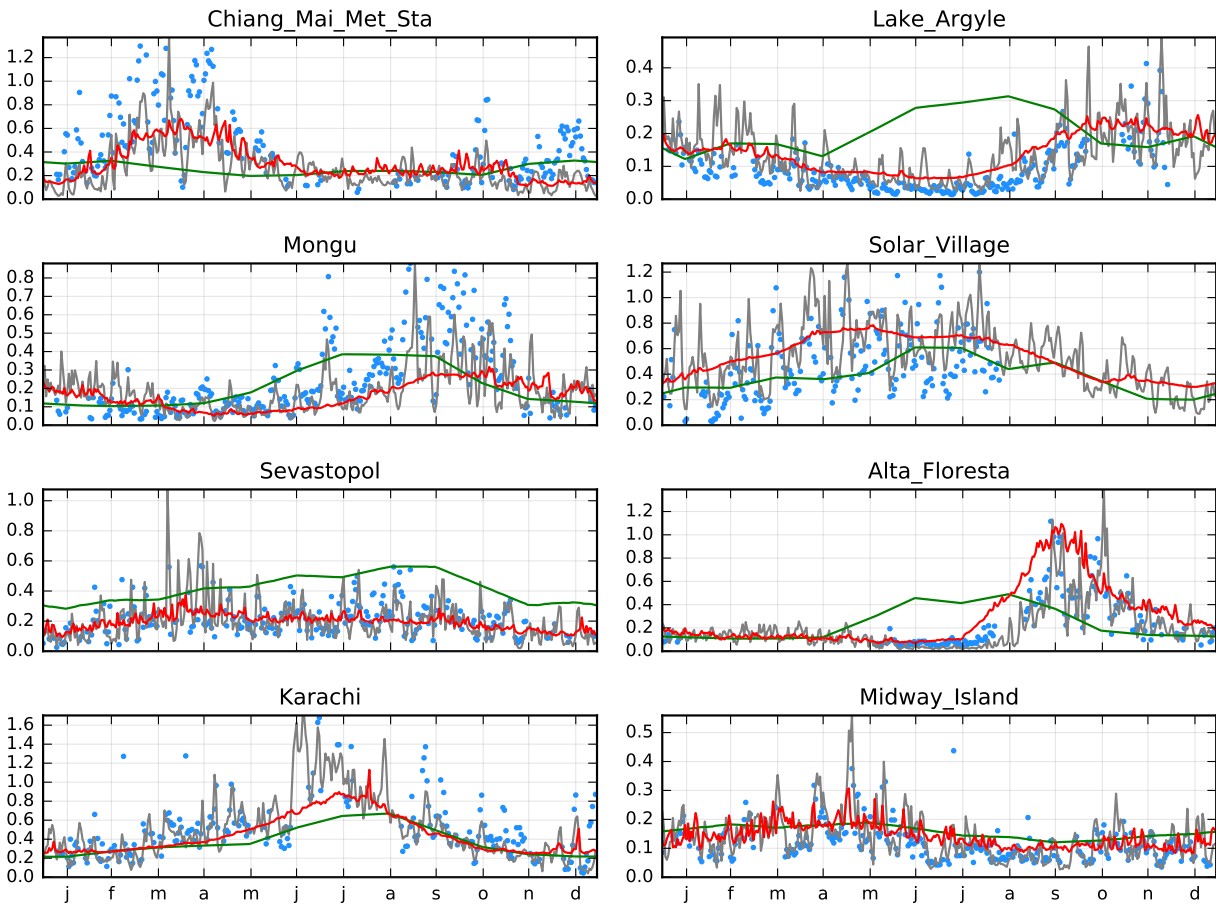

**Figure 7.** Comparison of daily mean total AOT at 500 nm for the year 2008 computed from the climatology presented in this work (red line) , from the TG97 climatology (green line) and from the CAMS Interim reanalysis prognostic fields (grey line) at eight AERONET sites. The AERONET L2.0 daily mean observations are shown as blue dots. Please notice the different Y axis for each plot.

All experiments are performed using the ECMWF model version 43R3 (operational from July 2017) with 137 vertical levels and prescribed sea surface temperature. We found that the results do not depend on the model horizontal resolution as long as this is high enough to represent well sub-continental atmospheric circulations. This is because, as we will see in the following, the climatological aerosol radiative effect acts on broad areas with largest impact on the bulk features of regional circulations. Therefore we adopted for these experiments an horizontal cubic-octahedral grid at an equivalent resolution of 0.2 degrees for efficiency. The CAMSiRA climatology was implemented with an horizontal resolution of 3x3 degrees and the native vertical grid of 60 levels. The fields are interpolated on-line to the horizontal grid used in the radiation scheme (Hogan and Bozzo, 2018). This resolution allows minimal impact on the model I/O while still being able to resolve the regional AOT features. Only the direct radiative effect of aerosol is taken into account in the IFS with no attempt at representing indirect effects on the cloud droplet concentration and effective size.

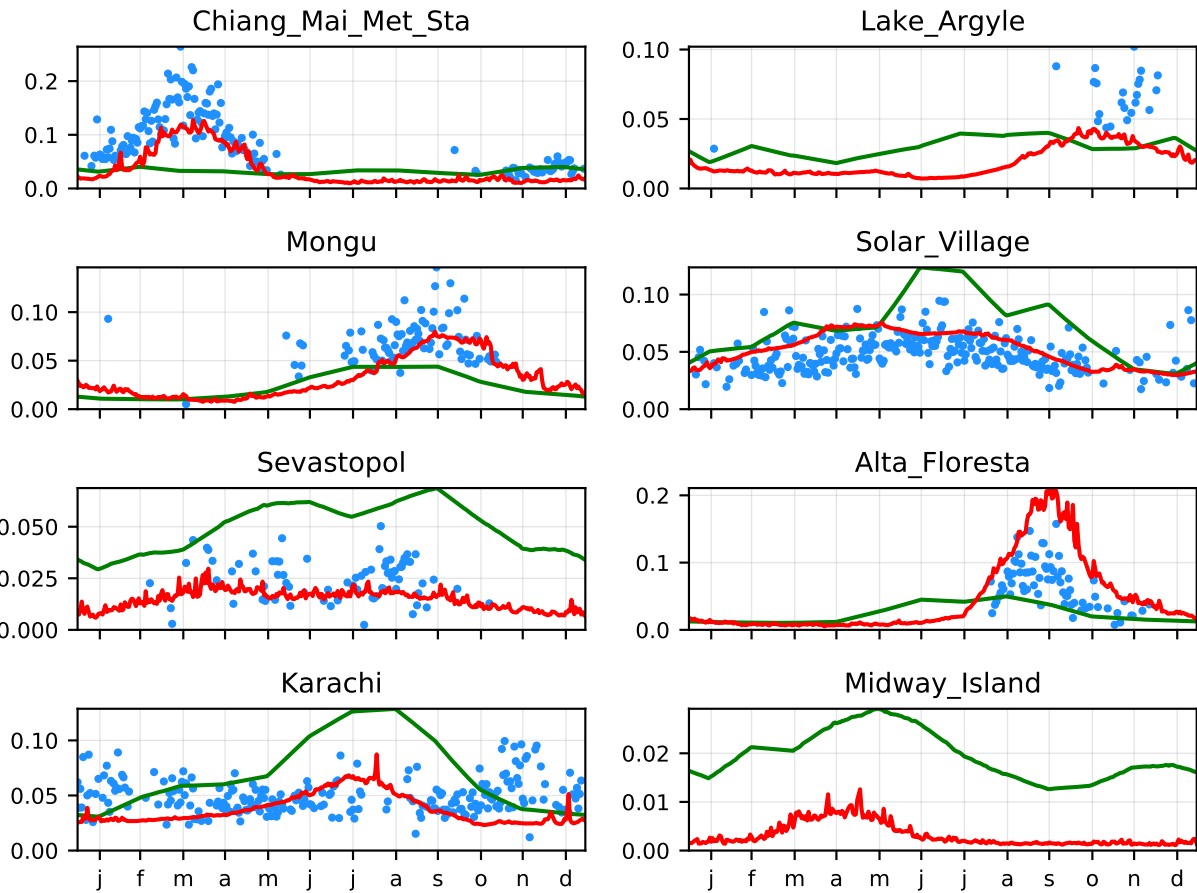

**Figure 8.** Comparison of daily mean AAOT at 500 nm from the CAMS climatology (red line), the TG97 climatology (green line) both as implemented in the ECMWF IFS (see appendix B for details) and AERONET data (blue dots). Aeronet data are from L2.0, version 2 retrieval products (Holben et al., 2006; Dubovik et al., 2002) averaged over 5 years between 2006 and 2010. Please notice the different Y axis for each plot.

We assess the model changes using two type of experiments: changes in mean radiative fluxes are assessed using a small ensemble of four one-year runs over the period 2001-2004 where the IFS is left running unconstrained for a full year. We will refer to these experiments as "climate runs".

The impact of changes in the aerosol climatology on the forecast skill is measured with a set of 10-day forecasts separated by 24 hours over the periods May-August and December-February 2016, run twice, once with the new CAMSiRA climatology and once the older TG79 climatology. Each forecast is initialized from the operational ECMWF analysis at 00 UTC and verified against the operational analysis. We will call these experiments "forecast runs".

## 4.2 Impact on radiative fluxes

The change from the TG97 to the CAMSiRA climatology affects radiative fluxes in the long-wave (LW) and short-wave (SW) both via direct interaction between aerosol and radiation and indirectly via changes in the clouds distribution due to diabatic forcing on temperature. Using the "climate runs" we find that at the surface the net SW flux decreases in places by up to 20-30 $W/m^2$ in the larger AOT in the areas of large organic species emissions and over the deserts (Fig 9 bottom). Although the large amount of sea salt aerosols over the oceans also contributes to a reduction of the surface net clear-sky SW radiation (by about 2-4 $W/m^2$) this is less significant given the larger contribution of cloud cover in those regions.

At the top of the atmosphere (TOA) the CAMS climatology increases the clear-sky reflected SW radiation globally with respect to TG97 by about 1-2 $W/m^2$ (not shown) but the effect on global all-sky fluxes is small and the largest impacts are confined to small areas affected by large dust plumes transported from the Sahara desert over the Atlantic Ocean. Changes observed over the Indian Ocean and South-East Asia (see Fig. 9 third row) depend mainly on local changes in the cloud distribution associated with aerosol-induced differences in regional atmospheric circulation. This aspect will be discussed in section 4.4.

The impact on the LW fluxes is small but it is significant in the regions with the largest mineral dust AOT, especially in the Northern Hemisphere summer months. The implementation of the CAMS climatology with the optical properties described in the Appendix B brings in our tests an increase of down-welling LW radiation at the surface which reaches more than 10 $W/m^2$ over the Western Sahara and Saudi Arabia. This in part offsets the reduction in incoming SW radiation at surface in those areas, which is of the order of $10 - 15$ $W/m^2$. At the TOA significant differences in LW fluxes between the two climatologies are found only where clear-sky dominates. Over the deserts high-level dust layers reduce the upward emission to space and the effect is stronger for the CAMS climatology which has a larger amount of dust mass over the deserts.

Most of the bias in TOA fluxes computed from the IFS operational cycle 43R3 against the top of atmosphere fluxes provided by the CERES-EBAF project (Clouds and the Earth's Radiant Energy System Energy Balanced and Filled, https://ceres.larc.nasa.gov/produc TOA last accessed 31 October 2019) are due to errors in cloud cover and cloud amount (e.g. Ahlgrimm et al., 2018) and aerosols only play a small role (first two rows in Fig 9). Therefore the modification in the aerosol distribution brings only small local improvements. In particular the change in the dust distribution over the Western Sahara and Central-East Atlantic brings some local reduction in both LW and SW TOA flux bias by increasing by about $5 - 10$ $W/m^2$ the reflected SW radiation and slightly decreasing by about 5 $W/m^2$ the out-going LW radiation. Some further small changes in TOA fluxes are observed over the Indian Ocean, but this time not related to changes in the clear sky radiation. Instead, a slight reduction in the cloud cover on the western coast of India indirectly improves the TOA fluxes by reducing the reflected SW radiation and at the same time increasing the outgoing LW radiation. This change in the total cloud cover in the area is related to changes in the aerosol radiative forcing which modifies the Summer monsoon circulation, as discussed in section 4.4.

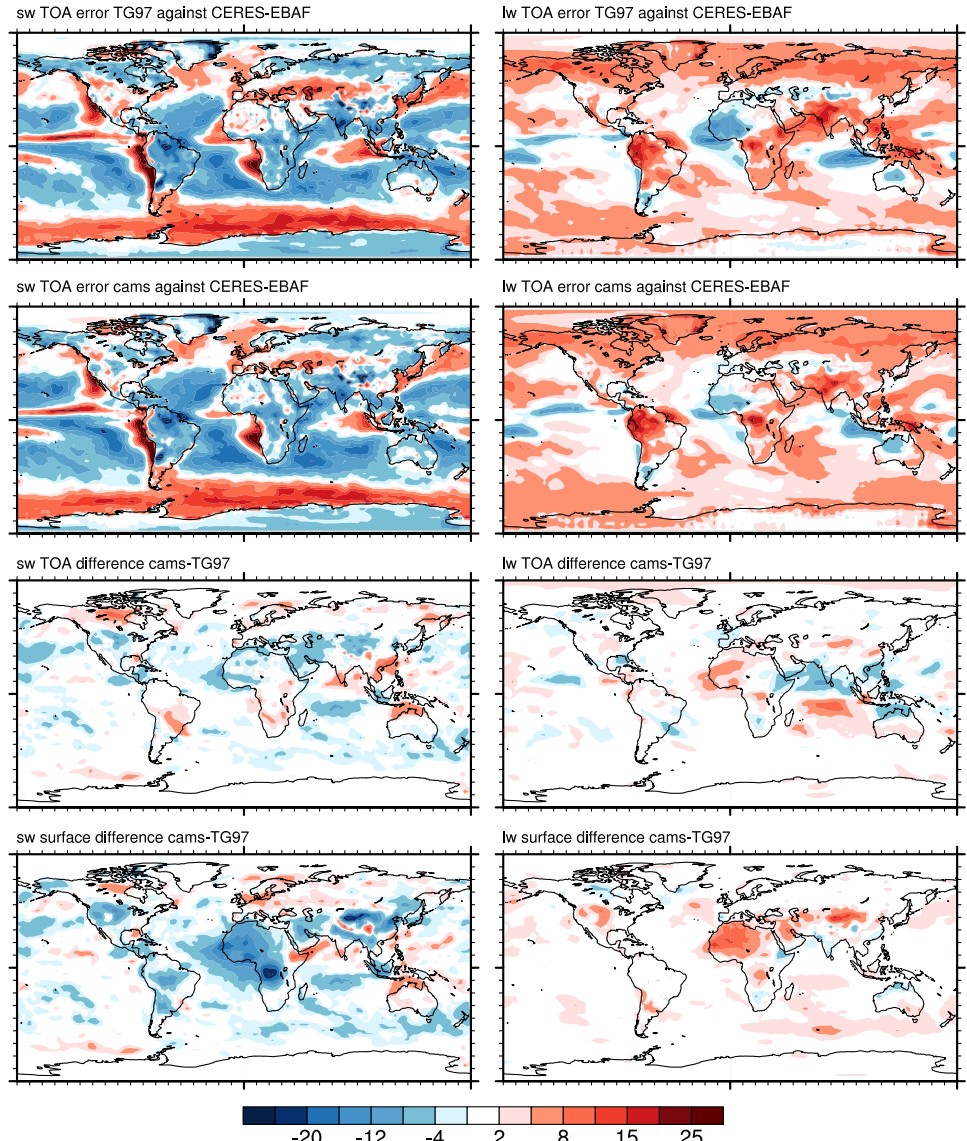

**Figure 9.** Changes in multi-annual mean (2001-2004) net radiative fluxes (in $W/m^2$) at the top of atmosphere (TOA) and surface for the short-wave (left column) and long-wave (right column) in the IFS. The top two rows show the errors in the TOA fluxes respectively when using TG97 and CAMS climatologies compared to CERES-EBAF observations. The last two rows show the change in the TOA and surface fluxes between the experiment using the CAMS climatology and the experiment using the TG97 climatology. Values above(below) 4(-4) $W/m^2$ indicate systematic features larger than the natural variability of the field.

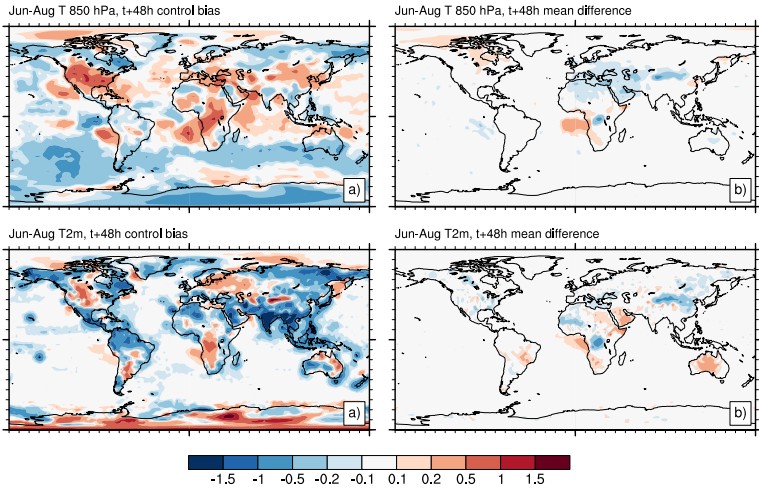

**Figure 10.** Temperature at 850 hPa (K, top row) and 2 m (K, bottom row) for forecast time t+48 hours averaged during the two months period July-August 2016. The left column shows the error of the operational model, the right column shows the difference between a forecast experiment with CAMS climatology and the operational model using TG97. Model errors are computed against operational analysis.

### 4.3  Impact on forecast errors and skill

Using the "forecast runs" we can measure to what extent changes in the direct aerosol radiative effect affect measures of forecast skill scores. The direct impact of a new aerosol climatology is to alter the radiative heating rate profiles and the surface energy budget. The former dominates the change in forecasts errors, and it affects the mid-to-lower tropospheric temperatures. The latter impacts mostly the surface temperature. Although the change in AOT is spatially highly in-homogeneous and locally large, this does not appear to be enough to impact the variability of the large-scale circulation.

Two main regions show the largest impact on the lower tropospheric temperature, and these are dominated by dust and organic/black carbon aerosols. Because of the combination of the difference in total AOT and in optical properties, for the same AOT the CAMSiRA climatology reduces the absorption of SW radiation in dust-affected regions with respect to TG97. This induces a widespread decrease in temperature of about 0.1 K below 700 hPa after 48h (Fig 10) growing to more than 0.2 K at day five. In the ECMWF model this helps reducing by about a third the positive temperature bias observed in the Mediterranean region and the Middle-East. The effect is larger in the summer months due to the stronger mean solar radiation and the larger dust AOT.

Another significant temperature change is observed below 850 hPa over the Gulf of Guinea and Central Africa where the relatively large amount of biomass burning aerosol in the CAMSiRA climatology significantly absorbs SW radiation. In this case the ECMWF model already suffers of a positive temperature bias in the region and the extra heating provided by the new aerosol AOT further increases the pre-existing bias. The radiative impact of the organic and black carbon species have

generally a small but positive impact on the upper-air temperature biases over Northern Canada in the summer months, although in these areas the seasonal variability of the forest fires is naturally impossible to capture in a climatological distribution. Single events can be very significant and they need prognostic treatment to accurately take into account their impact on local weather parameters (Toll et al., 2015).

Surface temperatures are affected by the change in aerosol climatology only locally over Central and North Africa and part of Asia (Fig 10), where changes in the AOT between the two climatologies is the largest (see Fig 4 ). In the biomass burning regions of Central Africa the decrease in surface SW radiation causes a decrease in the surface temperature which helps reducing the pre-existing positive bias. In North Africa and Middle-East the change in dust AOT is significant in the summer months, with surface cooling over the West Sahara and localized surface warming over Saudi Arabia where the significant

increase in down-welling LW compensates the smaller decrease in down-welling SW. Other significant temperature changes are found in Australia, where the reduction in dust AOT in the CAMSiRA climatology causes surface warming and in the Taklamakan desert in China, where the large dust AOT causes surface cooling.

The variability and forecast skill of large-scale extra-tropical weather patterns are not significantly affected by these regional changes in temperature. Measures of the forecast skill such as the anomaly correlation of mid-tropospheric geopotential, show

virtually no impact (not shown), corroborating similar results of Morcrette et al. (2011).

Measurable impacts on hemispheric scores are found only for the temperature RMSE in the lower troposphere during NH summer (Fig 11), due to the aforementioned changes in SW absorption by mineral dust and organic species. In particular, the temperature RMSE generally improves in the Northern Hemisphere by about $1\%$ in summer thanks to less SW absorption over the deserts which reduces the persistent warm bias affecting the IFS in the Northern Hemispheric Summer between the surface

and 700 hPa.

In the Tropics and partly in the Southern Hemisphere we found an increase by $\sim 1\% - 0.5\%$ in the 850 hPa temperature RMSE relative to the RMSE using TG97. This is dominated by the localized increase in forecast errors over the Gulf of Guinea (local increase in the RMSE up to 20%) which in turn affects the forecast skills over the tropical belt.

The scarce availability of continuous observations in this area makes it difficult to have a good estimate of the real aerosol

radiative effect, but the impact on the "forecast runs" suggests a bias in the CAMSiRA aerosols over the Gulf of Guinea given the significant departure from the temperature profile of the operational analysis. Independent estimates of AAOT for Central Africa (e.g. Bond et al., 2013) seem indeed to suggest a large overestimation in the CAMS model in the summer months over Central Africa. We identify as likely contributors to this bias (i) possible incorrect vertical distribution of all aerosols (including absorbing types) in the forecast model driven in part by an incorrect weight of the contribution by convective transport and

scavenging, (ii) a tendency for the assimilation to assign far too much relative importance to black carbon in the tropics over the other species, (iii) errors in the emission for organic and black carbon aerosols and (iv) too large absorption in the optical properties associated to the organic and black carbon species. These problems are currently being addressed and improvements have been incorporated in the most recent CAMS reanalysis (Inness et al., 2019).

This analysis shows that although the impact of a change in the aerosol climatology are small in terms of large-scale forecast

skill scores, nevertheless they can be not-negligible, especially in areas where the model has pre-existing biases or where the

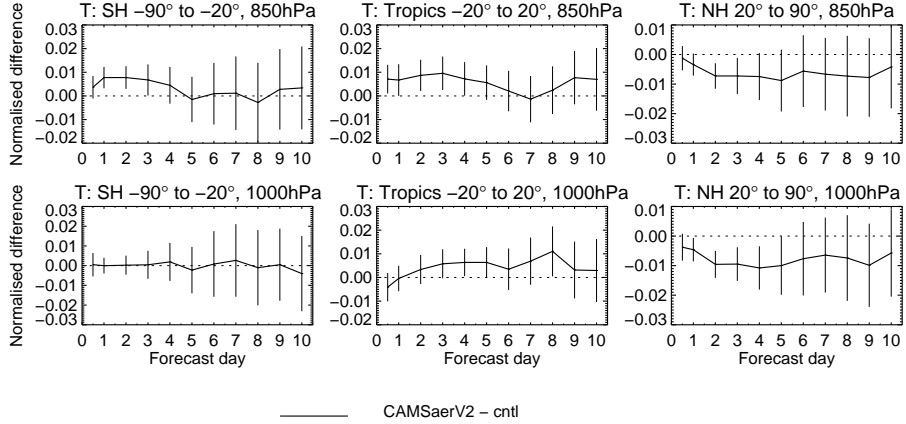

**Figure 11.** Normalized temperature RMSE difference at 1000 hPa and 850 hPa for a set of forecasts runs using the new CAMS climatology against the operational configuration. The experiments cover a summer season (2-May-2016 to 13-Aug-2016) and are verified against the operational analysis. Confidence range 95% with AR(2) inflation and Sidak correction for 4 independent tests (Geer, 2016). These experiments were done using a cubic-octahedral spectral truncation TCo399 but the main results are independent on the model resolution. Values $< 0$ mean that the forecasts with the CAMS climatology are better than those with the TG97 climatology.

model has particularly low errors and is therefore sensitive to small changes in the local radiation budget. On the other hand, impacts at regional scale can be large. The most robust changes are found over the Indian Ocean during the summer monsoon season and are forced predominantly by a modified radiative forcing by the desert dust which brings a reduction in the near-surface wind errors. Section 4.4 presents in more details the feed-backs between the monsoon circulation and changes to the local AOT.

## 4.4 Impacts on local circulations: the summer Indian monsoon

The area of the Northern Indian Ocean during the summer monsoon season shows the largest feedback between changes in aerosol radiative forcing and regional-scale circulation. In this region the CAMS climatology has a different impact on radiative fluxes than the TG97 climatology as implemented in the IFS (Fig 4 and Fig 9). The largest change occurring with the CAMS climatology during summer is a decrease in total SW absorption over the Middle-East and East Africa of approximately 4-8 $W/m^2$ on average, but exceeding 30 $W/m^2$ over the Horn of Africa. This is due to both a change in the distribution of mineral dust mass in the region and to the higher dust reflectivity we adopted in the CAMS climatology.

Numerous studies have explored the sensitivity of the summer Indian monsoon to aerosol radiative forcing from both anthropogenic and natural sources (Bollasina et al., 2011; Lau and Kim, 2006; Wang et al., 2009). By using a combination of model and satellite data Vinoj et al. (2014) showed that the radiative effect of mineral dust over Eastern Africa and Arabian Peninsula affects the monsoon circulation over the Indian Ocean. The heating rate perturbation induced by the dust layer can modulate

the strength of low level westerly zonal winds and moisture transport towards Eastern and Central India over time scales of weeks. The feedback was successively explored in detail by Jin et al. (2015) and Jin et al. (2016) showing the thermodynamic mechanism that links the dust radiative effect over the Iranian Plateau and precipitation variability on sub-seasonal time scales over western India. This implies that a realistic representation of the aerosol radiative effect in the region can potentially have

a significant impact on the predictability of the monsoon circulation in medium-range and seasonal forecasts.

In the operational configuration pre-CY43R3 with un-coupled sea-surface temperatures, the IFS has a too strong near-surface westerly jet across the northern Indian ocean, from the Eastern Africa to the Western India (Fig 12a) which in turns causes too wet conditions over Western India during the summer months. This brings a positive precipitation bias in the region of 1-2 mm/day over land in the three month period June-August as compared to various estimates of surface precipitation (Table 3).

The same circulation bias is also responsible for part of the errors in the top-of-atmosphere LW and SW fluxes (Fig 9) observed in the "climate runs", related to too much cloudiness over Western India.

**Table 3.** Mean precipitation over Western India (region boundaries: lat 25N-6N;lon 67E-77E) for JJA estimated by different products and model bias for two forecast experiments for the period 2001 to 2004. Data are in mm/day.

| | GPCP v2.2[1] | HOAPS3 v6*[2] | SSMI*[3] | TRMM 3B43[4] |
|---|---|---|---|---|
| OBS | 5.5 | 5.1 | 2.5 | 6.4 |
| TG97-OBS | 2.1 | 0.2 | 4.1 | 1.3 |
| CAMS-OBS | 1.4 | -0.3 | 3.8 | 0.5 |

*values not defined on land grid points

(1)Global Precipitation Climatology Project, (Huffman et al., 2015)

(2)Hamburg Ocean Atmosphere Parameters and Fluxes from Satellite Data, (Andersson et al., 2010)

(3)Special Sensor Microwave/Imager and Sounder, (Wentz et al., 2012)

(4)Tropical Rainfall Measuring Mission Multi-Satellite Precipitation Analysis, (Huffman et al., 2007, 2010)

The CAMS climatology brings changes in the mean winds and temperature below 700 hPa and this reduces the forecasts errors in the area both for the wind strength at all lead times (Fig 12b) and also for the accumulated seasonal errors in precipitation amounts (Table 3). Further evidence of an improved mean model state come also from the assimilation cycle. The increase

in surface temperature and pressure over the Persian Gulf and Saudi Arabia helps reducing the error of the forecast first-guess with respect to the observations used in the assimilation step, indicating an improvement in the analysis fields because the model is closer to the observations (not shown).

Near-surface westerly zonal wind strength decreases in the northern part of the Indian Ocean and increases to the south (Fig 12b), implying a weakening and southward shift of the low-level jet. The changes grow larger at longer lead times due to

the cumulative contribution of the modified radiative forcing acting from the very beginning of the forecast. These circulation changes are the result of a combination of large-scale and more localized perturbations to the temperature gradients between the Indian Ocean and the land areas.

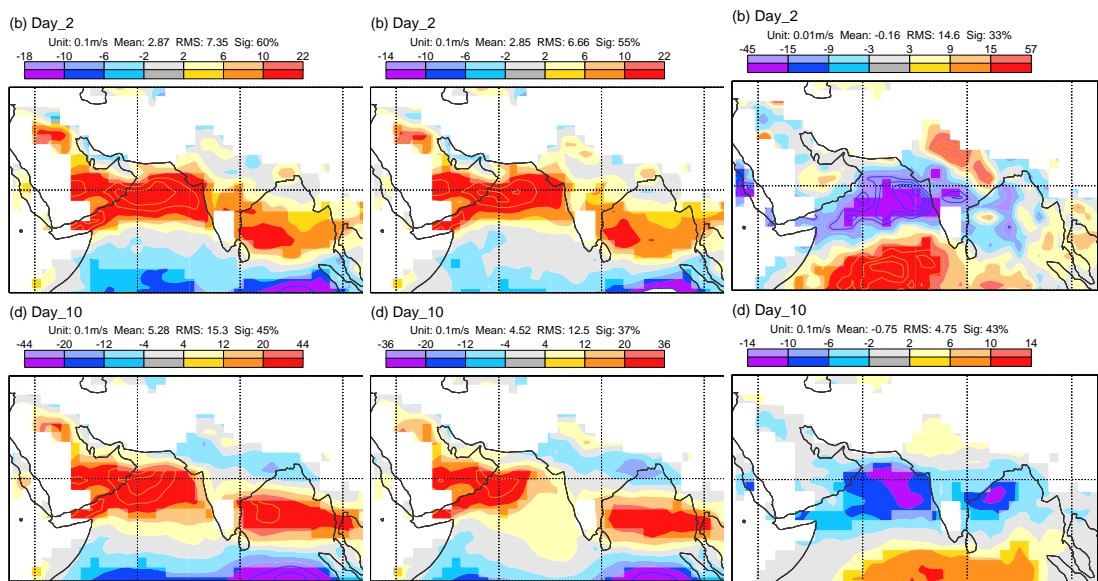

**Figure 12.** Near surface (925 hPa) zonal wind for the period 1st of May 21st of August 2016 over the northern Indian Ocean for forecast day 2 (top row) and forecast day 10 (bottom row). Model bias using TG97 (left), CAMS climatology (middle) and difference between forecasts using the CAMS climatology and TG97 climatology (right). Notice the different scale in the right-hand side panels. Bold colors indicate areas significant at the 5% level using a paired T-test with AR(1) noise. The units are indicated above each figure.

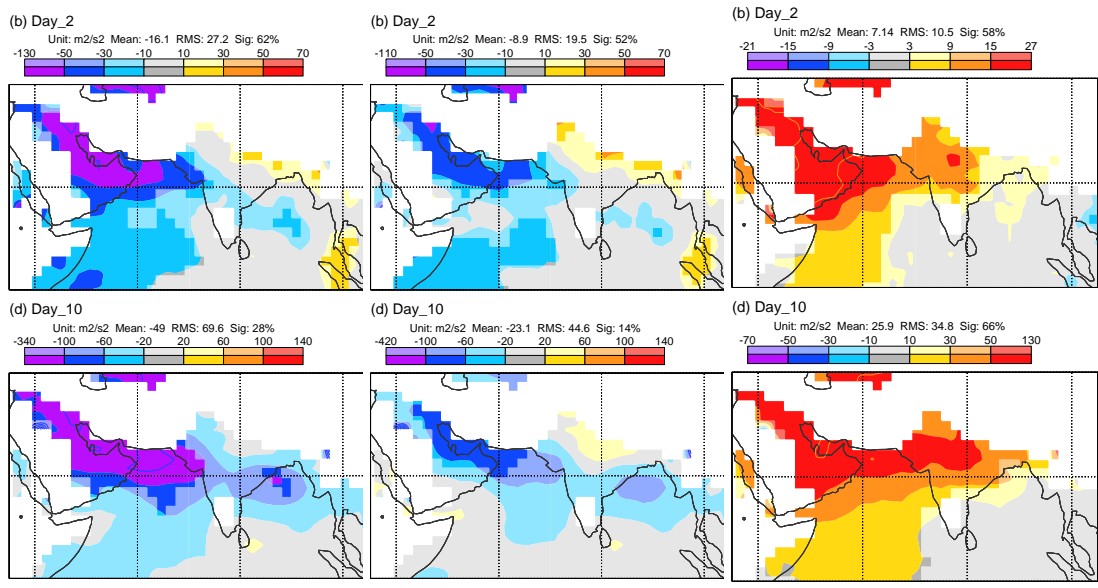

**Figure 13.** As Fig. 12 but for the geopotential at 925 hPa ($m^2/s^2$).

In the CAMS climatology less SW radiation is absorbed by the dust layer causing a decrease in the lower tropospheric temperature over the Eastern Africa/Arabic peninsula region (see Fig. 10), a key driver of the monsoon circulation over the Indian Ocean (Vinoj et al., 2014; Jin et al., 2016, 2015). Following this lower tropospheric cooling, the geopotential height decreases over land above 500 hPa inducing upper level convergence and localized descending motion which partially balances the radiative cooling. This causes an increase in the surface pressure and geopotential height at low levels over the Middle-East and Arabian peninsula, improving the model bias by up to 30%-50% (Fig 13). The higher pressure below 800 hPa reduces the low-level convergent flow over the continental areas resulting in a weaker north-eastward circulation in the northern section of the Indian Ocean.

Moreover, although in the IFS aerosol concentration do not directly impact cloud microphysics, yet the changes in the local atmospheric circulation and the vertical distribution of heating rates can cause an indirect impact of aerosol on cloudiness. Over the Indian Ocean the weaker monsoon circulation implies a reduced average cloudiness with a clear impact in the radiative fluxes at the top of atmosphere, as observed in section 4.2.

We tested how much these effects depend on the absorption properties of dust using the optical properties computed from different the refractive indices such as Dubovik et al. (2002), with less absorption at shorter wavelengths (see Table A3), and we found that very weak absorbing dust produces an even stronger decrease in the monsoon circulation (not shown), confirming the previous findings (Vinoj et al., 2014). This sensitivity, together with the fact that in our experiments the SST are prescribed and that we do not explicitly simulate the interaction between aerosol and cloud microphysics, indicates that the direct atmospheric heating by the dust layer is the main factor behind the observed circulation changes.

## 5   Conclusions

This work documents a new monthly-mean climatology of aerosol distribution based on the Interim reanalysis from CAMS (CAMSiRA, Flemming et al., 2017) and its control run. The data set represents a monthly-mean distribution of mass mixing ratio of 5 aerosol species sub-divided into 11 types over 60 vertical levels. The climatology is available at full native resolution for the 3-dimensional fields or at any coarser horizontal grid. The user can associate the radiative properties of choice to the aerosol distribution, and here we present results used the bulk properties for each species computed for the 30 radiative bands of the ECMWF radiative scheme (Hogan and Bozzo, 2018).

We tested the impact of the CAMS climatology on the ECMWF Integrated Forecasting System in comparison to the aerosol climatology operational until Summer 2017 which was derived from Tegen et al. (1997).

Compared to AERONET observations over number of years, the CAMSiRA climatology captures fairly well the mean seasonal variation of the total AOT while the dependence of the optical properties on the relative humidity helps capturing at least part of the daily variability. We used the AAOT retrieved from AERONET sites to test the AAOT resulting from associating the optical properties used in the radiation scheme of the ECMWF IFS to the present climatology. The comparison showed generally a good agreement, but it did also highlight the uncertainty in the definition of the single scattering albedo over various regions, in particular when dominated by biomass burning events. We did not attempt a thorough discussion of

the variaty of refractive indices to associate to the various species and the user has the opportunity to experiment with different optical properties.

When implemented in the ECMWF IFS, the new CAMSiRA aerosol climatology affects the radiative fluxes and brings small improvements locally to biases both in the short-wave and in the long-wave spectrum compared to satellite observations. These changes in aerosol radiative forcing with respect to the ECMWF implementation of the Tegen et al. (1997) climatology are due to a different spatial distribution, different radiative properties and different representation of the size distribution of each aerosol species.

In the ECMWF IFS the change in the climatological representation of aerosol distribution has a limited impact on commonly used measures of hemispheric forecast skill scores and it does not affect significantly the variability of the large-scale synoptic circulation, in agreement with recent studies (e.g. Morcrette et al., 2011; Mulcahy et al., 2014; Toll et al., 2016). Locally, temperature changes in the lower troposphere can be of similar order of magnitude of the pre-existing model biases and and can therefore affect the model mean state.

Larger impacts are found in areas where there is a stronger link between clear-sky radiative perturbations and local circulation. In the ECMWF model the summer Indian monsoon over the Indian Ocean shows a marked sensitivity to the mineral dust radiative forcing over Eastern Africa and Saudi Arabia. The magnitude of the absorption by mineral dust modifies the mean temperature and geopotential over the Middle-East land areas and the Indian Ocean which in turn affects the north-eastward branch of the the Indian monsoon. In the ECMWF model this reduces by about 30% the mean model bias in lower-tropospheric geopotential and zonal wind, also improving the representation of precipitation over the northern Indian Ocean and south-west India.

The CAMS prognostic aerosol model is in continuous development and future releases of this aerosol climatology will incorporate the latest improvements in aerosol modelling and data assimilation.

## 6 Code availability

The IFS source code is available subject to a licence agreement with ECMWF; see also Flemming et al. (2015); Rémy et al. (2019) for details. The code used to generate the optical properties for each aerosol species is based on the standard Wiscombe (1980) scheme for Mie scattering.

## 7 Data availability

Two datasets described in this work are available from the CAMS data repository: the monthly mean layer-integrated mass mixing ratio of all aerosol species at a resolution of 3x3 degrees and 60 vertical levels and the optical properties computed for each species for the 30 spectral band of the ECMWF radiation code (Hogan and Bozzo, 2018). The data are hosted on the CAMS data archive and available for download at https://doi.org/10.24380/jgs8-sc58. A Confluence web knowloedge

page can be found at the following web address: https://confluence.ecmwf.int/display/CKB/CAMS+Monthly+Mean+Aerosol+Climatology+for+global+models.

*Author contributions.* ABo prepared the database, analyzed the experiments and drafted this document. JF, ZK, SR, ABe developed and improved the CAMS aerosol model and helped running and analyzing the experiments. All authors contributed to the final version of the document.

*Competing interests.* The authors declare no competing interests.

*Acknowledgements.* We thank Irina Sandu, Robin Hogan, Richard Forbes, Mark Rodwell and Julien Chimot for their valuable comments at various stages of this work. We also thank the CAMS consortium for their work towards the improvement of the atmospheric composition modules coupled to the IFS and for their efforts in the evaluation of the atmospheric composition products. Luke Jones provided much needed support in for the AERONET comparisons. We thank the PIs and Co-Is and their staff for establishing and maintaining the AERONET sites used in this investigation. Two anonymous reviewers provided very helpful comments which contributed significantly to the improvement of the initial version of this manuscript.

## Appendix A:  Vertical scale height of the CAMS aerosol types

We can derive an estimate of the scale height $\xi$ for each aerosol type from the vertical distribution of the mass mixing ratio in the CR. Generally $\xi$ depends on the aerosol spatial distribution and the season and it can be found by calculating at every grid point the height at which the normalized cumulative mass distribution reaches the value $1/e$. The spatial distribution of $\xi$ is shown in Fig A3 for July and January for all aerosol types.

The scale height of mineral dust exhibits the largest spatial and seasonal variations because of the strong dependence of the dust emission and transport on the height of the boundary layer mixing over the deserts and the seasonal patterns of large-scale synoptic circulations. The dust species is the only one exhibiting a large seasonal cycle, with $\xi$ ranging from $\sim 2$ km in winter rising to $\sim 3$ km in summer. For the other species $\xi$ can be approximated by a constant value throughout the year. Sea salt aerosols and black carbon are generally confined to the lower levels with $\xi \sim 1$ km, while the organic matter extends higher with $\xi \sim 2$ km. As sulfate is formed from $SO_2$, which has sources from both anthropogenic activities and from oceanic dimethyl sulfate, it occurs further over most of oceans and continents and tends to have a more homogeneous distribution with $\xi \sim 3$ km (Fig A3).

The scale height can be used to distribute vertically the species in case only a two-dimensional distribution of total mass needs to be used with less accurate vertical distribution. The older implementation of the TG97 climatology in the IFS used a simple pressure-based exponential function of type

$$(p/p_0)^{(H/\xi)}, \tag{A1}$$

with $p_0$ pressure at the lowest model level, $H = 8.4$ km the scale height of the standard atmosphere and $\xi$ the scale height of the aerosol component. The most significant contribution to bias related to a less accurate description of the aerosol vertical profile is expected from the absorbing species which determine the vertical profile of short-wave heating rate.

## Appendix B:  Optical properties

Here we briefly describe the set we used in the IFS implementation described in this work. The user can customize the the optical properties to associate to the aerosol climatology according to the specific needs of the application, but any large departure from the extinction coefficients described here will reflect in a change to the total AOT from the one obtained in CAMSiRA. The aerosol optical properties are computed for each of the 14 short-wave (SW) and 16 long-wave (LW) bands of the RRTM (cite AER inc.) radiation scheme on which the IFS radiation scheme is based (ECRAD, Hogan and Bozzo, 2018). Spherical shape is assumed for all species, with a number size distribution described by a log-normal function similar to the original version of the aerosol scheme (Reddy et al., 2005) and defined as:

$$n(r) = \frac{dN(r)}{dr} = \frac{N}{\sqrt{2\pi}r\ln(\sigma)} exp\left(-\frac{\ln^2(r/r_{mod})}{2\ln^2(\sigma)}\right) \tag{B1}$$

with $N$ total particle number concentration, $\sigma$ geometric standard deviation and $r_{mod}$ mode radius.

Table A1 lists the relevant parameters of the distribution for each species. The bulk optical properties (mass extinction coefficient, single scattering albedo ($\omega$) and asymmetry parameter ($g$)) are computed with a standard code for Mie scattering based on Wiscombe (1980). For the hydrophilic types the optical properties change with the relative humidity due to the swelling of the water soluble component in wetter environments. The refractive index ($m$) and density ($\rho$) of the aerosol particle change according to the relations (Koepke et al., 1997):

$$\rho = \rho_{dry} * r_{dry}^3/r^3 + \rho_{water} * (r^3 - r_{dry}^3)/r^3 \tag{B2a}$$

$$m = m_{water} + (m_{dry} - m_{water}) * r_{dry}^3/r^3 \tag{B2b}$$

with $r_{dry}$ and $r$ the mode radius respectively of the dry particle and at a relative humidity value. The size distribution is modified applying growth factors (Table A2) to the mode radius and to the limits of integration, maintaining the same geometric standard deviation. The mass mixing ratio in the climatology is defined for the dry mass for sulfates and organic matter but for a mass relative to 80 % relative humidity for sea salt. The optical properties are computed taking this into account.

A brief description of the refractive index associated to each aerosol type is given in the following paragraphs.

**Organic matter:** The optical properties are based on the "continental" mixtures described in Hess et al. (1998). The mixture represents aerosols over continental areas influenced by anthropogenic and natural emissions. We used a combination of 13% in mass of insoluble soil and organic particles, 84% of water soluble particles originated from gas to particle conversion containing sulfates, nitrates and organic substances and a 3% of soot particles. The combination gives optical properties representing an average of biomass and anthropogenic organic carbon aerosols. The refractive indices and the parameters used in the particle size distribution of each component are as described in Hess et al. (1998). The hydrophobic organic matter type uses the same set of optical properties but for a fixed relative humidity of 20%.

**Black carbon:** The refractive index used in the Mie computations is based on the OPAC SOOT model. At the moment the hydrophilic type of the black carbon species is not implemented and both types are treated as independent from the relative humidity. The single particle properties are integrated with a log-normal particle size distribution for sizes between 0.005 and 0.5 $\mu m$.

**Sulfate:** The sulfate type represents aerosol originated from sulfur emissions from industrial and fossil fuel combustion, biomass burning and natural sources (volcanic and biogenic). The refractive index is taken from the Global Aerosol Climatology Project (GACP, http://gacp.giss.nasa.gov/data_sets/) and it is representative of dry ammonium sulfate $(NH_4)2SO_4$. The hygroscopic growth is parameterized after Tang and Munkelwitz (1994) and reported in Table A2.

**Mineral dust:** The large uncertainty in mineral dust composition (e.g. Colarco et al., 2014) means that it is difficult to represent the radiative properties of this species with a single refractive index fitting different part of the World. We show here three choices spanning different SW absorption properties. Woodward (2001) combined measurements from different locations and provides the largest absorption in the visible range with an imaginary refractive index at 500 nm of $n_{i,500} = 0.0057$. Fouquart et al. (1987) propose a much smaller value $n_{i,500} = 0.0013$ and it represents the lower bound for mineral dust absorption. Dubovik et al. (2002) used AERONET measurements to retrieve the refractive index of mineral dust in different locations. For the Sahara region they report $n_{i,500} \sim 0.0022$ representing a value in between the previous two. The optical properties are computed individually for each of the three size intervals in the CAMS mineral dust model, using a log-normal size distribution with particle radius limits 0.03, 0.55, 0.9, 20 $\mu m$. For the IFS implementation described in this work we adopted Woodward (2001) as it resulted in the best overall impact on the IFS scores.

**Sea salt:** The refractive index for sea water is as in the OPAC database and the optical properties are integrated across the three size ranges in the CAMS model, using bi-modal lognormal distributions with particle radius limits 0.03, 0.5, 5, 20 $\mu m$ as in Reddy et al. (2005) and with the same hygroscopic factors according to Tang (1997), Table A2.

The complete set of bulk optical properties for all aerosol types, is shown in Figure A1 for the full range of spectral bands used in ECRAD. In Figure A2 the optical properties used in CAMSiRA climatology are compared to the properties used in the IFS for the TG97 climatology.

**Table A1.** Refractive index and parameters of the size distribution associated to each aerosol type in the CAMS model ($r_{mod}$ =mode radius, $\rho$=particle density, $\sigma$=geometric standard deviation). Values are for the dry aerosol a part from sea salt which is given at 80%RH. The organic matter type is represented by a mixture of three OPAC types similar to the average continental mixture, as described in Hess et al. (1998).

| aerosol type | size bin limits (sphere radius, $\mu m$) | Refr. index source | $\rho$ $(kg/m^3)$ | $r_{mod}$ $(\mu m)$ | $\sigma$ |
|---|---|---|---|---|---|
| Sea salt* (80% RH) | 0.03-0.5 0.5-5.0 5.0-20 | OPAC | 1.183e3 | 0.1992,1.992 | 1.9,2.0 |
| Dust | 0.03-0.55 0.55-0.9 0.9-20 | Dubovik et al. 2002 or Woodward et al. 2001 or Fouquart et al. 1987 | 2.61e3 | 0.29 | 2.0 |
| Black carbon | 0.005-0.5 | OPAC (SOOT) | 1.0e3 | 0.0118 | 2.0 |
| Sulfates | 0.005-20 | Lacis et al. (GACP) | 1.76e3 | 0.0355 | 2.0 |
| Organic matter[+] | 0.005-20 | WASO+ OPAC INSO+ SOOT | 1.8e3 2.0e3 1.0e3 | 0.0212 0.471 0.0118 | 2.24 2.51 2.00 |

*Sea salt is described by a bi-modal log-normal distribution with fixed number concentrations of 70 $cm^{-3}$ and 3 $cm^{-3}$ for the small and the large mode respectively.

[+]The species are mixed by number concentration. The individual number concentrations are 12000 $cm^{-3}$ (WASO), 0.1 $cm^{-3}$ (INSO), 8300 $cm^{-3}$ (SOOT) The hydrophobic component of organic matter uses the same optical properties but for a fixed relative humidity of 20%

**Table A2.** Growth factors used to characterize the size distributions of sea salt, sulfates and organic matter

| RH (%) | 0 | 10 | 20 | 30 | 40 | 50 | 60 | 70 | 80 | 85 | 90 | 95 |
|---|---|---|---|---|---|---|---|---|---|---|---|---|
| Sea salt | 1.0 | 1.0 | 1.0 | 1.0 | 1.44 | 1.55 | 1.666 | 1.799 | 1.988 | 2.131 | 2.36 | 2.877 |
| Sulfates | 1.0 | 1.0 | 1.0 | 1.0 | 1.169 | 1.220 | 1.282 | 1.363 | 1.485 | 1.581 | 1.732 | 2.085 |
| WASO | 1. | 1.05 | 1.09 | 1.14 | 1.19 | 1.24 | 1.29 | 1.34 | 1.44 | 1.54 | 1.64 | 1.88 |

growth factors for sea salt are from Tang (1997), growth factors for sulfates are from Tang and Munkelwitz (1994), growth factors for the OPAC species WASO are from Hess et al. (1998).

**Table A3.** Dust optical properties for the ECRAD band 400-700 nm computed using different refractive indices (mass extinction coefficient $k, m^2/g$, single scattering albedo $\omega$ and asymmetry parameter $g$). Data are for each of the three size bins of the CAMS aerosol model (bin limits in terms of particle radious: 0.03, 0.55, 0.9, 20 $\mu m$)

| RI | $k$ | $\omega$ | $g$ |
|---|---|---|---|
| Woodward (2001) | 2.5,0.95,0.4 | 0.96,0.90,0.83 | 0.68,0.67,0.80 |
| Dubovik et al. (2002) | 2.4,0.98,0.4 | 0.98,0.96,0.92 | 0.65,0.67,0.76 |

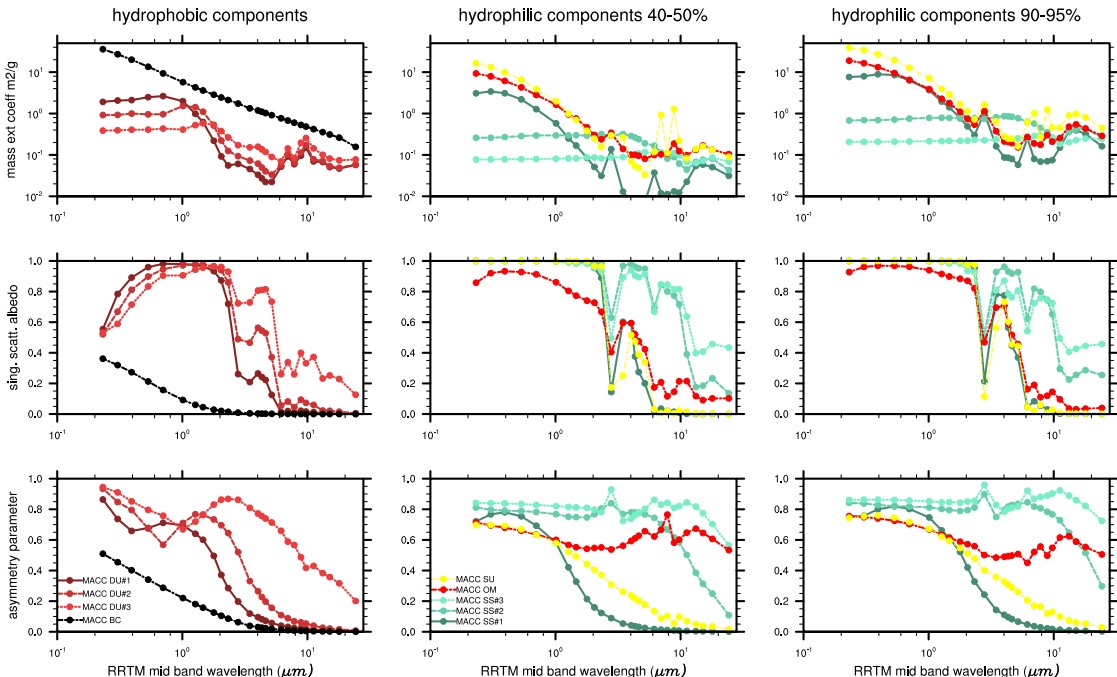

**Figure A1.** Optical properties of the aerosol species in the CAMS model for the 30 spectral bands of the ECMWF radiation scheme. For the hydrophilic species the mass extinction coefficient is computed with respect to the dry aerosol mass. The top row shows the mass extinction coefficient, the middle row shows the single scatter albedo and the bottom row shows the asymmetry parameter. The first column is for the hydrophobic species and the middle and right columns are for the hydrophilic species at two values of RH.

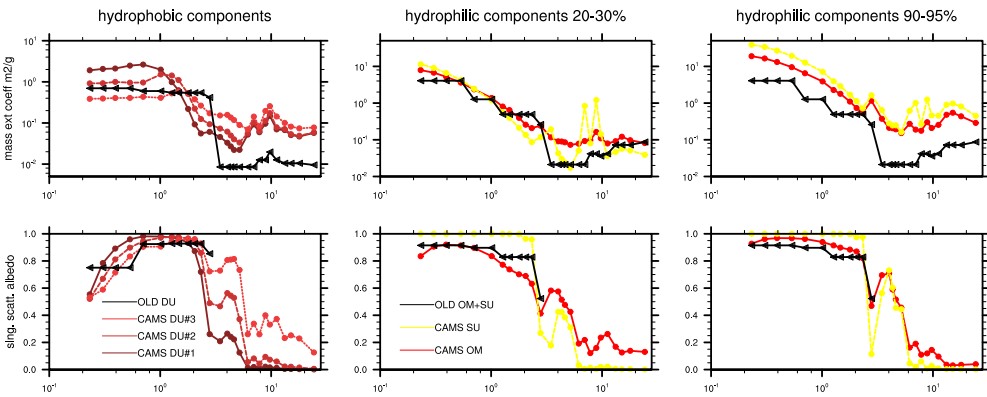

**Figure A2.** Comparison of optical properties used to describe the radiative effect of the aerosol species in the CAMS model (coloured lines) and in the old climatology based on TG97 (black lines). Values are for the 30 spectral bands of the ECMWF radiation scheme. For the hydrophilic species the mass extinction coefficient is computed with respect to the dry aerosol mass. The top row shows the mass extinction coefficient and the bottom row shows the single scatter albedo. The first column is for the hydrophobic species mineral dust and the middle and right columns are for the hydrophilic species organic matter and sulfates and for the CAMS climatology are shown at two values of relative humidity.

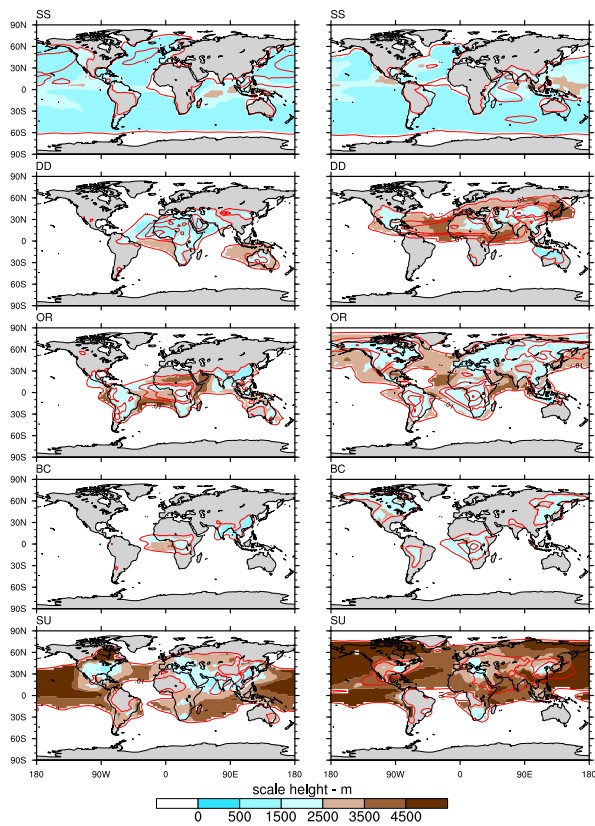

**Figure A3.** Scale height (color shade) and AOT (red contours) for each aerosol type for January (left) and July (right) computed from the CAMS Control Run over the years 2003-2013 and with the total AOT scaled to preserve the CAMS reanalysis total AOT. The scale height is shown only for the grid points with an AOT for that aerosol type larger than 0.01. Contour lines values are 0.01,0.05,0.1,0.4,0.8,1

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
