# Peer review of "An aerosol climatology for global models based on the tropospheric aerosol scheme in the Integrated Forecasting System of ECMWF."

_Geoscientific Model Development, 2019_

## Referee Comment (RC1) · Anonymous Referee #1 · 2 Aug 2019

General comments:

The manuscript is well written can be useful for potential users of the CAMSiRA aerosol climatology. In particular in showing the effects of the impacts of radiation fluxes using the new climatology in the IFS forecast system the effects of the new climatology are presented.

A major concern is that the aerosol climatology is only evaluated in terms of AOT. As the new aerosol climatology was constrained by MODIS AOT, it is nice to know but unsurprising that the new CAMSiRA climatology provides a better match with AOT measurements compared to the older Tegen et al. (1997) climatology given that the

older climatology was compiled from a results of very early attempts at aerosol tracer models using very coarse models and useemission fields that are meanwhile outdated. While it is a good start to look at regions that are dominated by specific aerosol types (although at most stations AOT will be a result of mixtures of different aerosol types, e.g. at Midway Island there is likely a contribution from sulfate AOT) it is notable that in particular for mineral dust evaluation at sites that are dominates by dust are absent, and should be added. As it is important for its radiative effect, particularly the effect on the Indian summer monsoon, the authors should also compare their absorbing AOT with the AERONET absorbing aerosol product. This should be a straightforward extension of the already existing analysis. Ultimately other aspects such as the mixing rations or number size distributions will be used (e.g. for simulating indirect effects of aerosol particles on clouds). Aerosol composition will play a major role for these aspects. How about comparing other aspects such as near surface concentrations?

Specific comments:

1. While the introduction section gives a detailed overview about the role of aerosol climatologies in NWP and in particular for the ECMWF model, to avoid confusions the section would benefit from a table listing the current and previous aerosol climatology versions.

2. Page 4, line 32: What does 'mass volumetric concentration' mean? Do you just mean 'mass concentration'?

3. Page 5, lines 21-25: This sentence is not clear, please explain in more detail what is meant by 'not efficient coupling' between convective transport and scavenging/speciation/vertical distribution of analysis increments.

4. Figure 1 The labels with numbers in some oft he panels (top 2 rows) are not explained. Are they actually needed?

5. Figure 1: In addition to mass load, the distribution of the AOTs of the individual

species would be interesting, as the AOTs ultimately determine the radiative effects. This would also support the choice of Aeronet locations relevant for individual aerosol types. These locations could be indicated on such AOT maps.

6. Figure 4: Additional difference plots between the two climatologies would be useful to highlight their key differences.

7. Page 11 and figure 5: At least one Aeronet station dominated by mineral dust should be added, as this aerosol type caused major differences between the climatologies.

8. Figure 5: what causes the dips in the green line (Tegen climatology) at the beginning of each month?

9. Page 14, lines 10-11: Please state here for which years the 'forecast runs' are performed. In the caption of Figure 8, the period May to August of the year 2016 is named, which should also be stated in the text.

10. Figures 6 and 7: Please provide the information on the years of the simulations in the figure captions

11. Figure 10: If, as stated in the figure caption, the figure shows also zonal winds as in Figure 9, why is the unit m2/s2 rather than m/s?

Minor corrections:

12. Abstract, line 1: 'global atmospheric models' – the words should not be starting with captital letters

13. Abstract, line 3: into -> in

14. Abstract, line 8: . . . assimilating -the- aerosol optical thickness. . .

15. The authors use at several places in the manuscript the expression 'specie' for singular of 'species'. Please check if that is the correct usage of the singular word here. (I am not a native speaker, but would also use species for singular and plural in

this context)

16. Page 6, figure 1 caption, line 1: Interim reanalysis is written as interim Reanalysis at other places in the manuscript, please make sure it is written with the same capitalization everywhere.

17. Page 7, line 29 ad -> and

18. Page 10, figure 4: I suggest to place the figure labels (a and b) above and not below the figures

19. Figure 5: The lines in figure and the labels are difficult to recognize. The lines should be thicker and the label fonts should be larger.

20. Page 17, line 10: fig -> Fig

21. Page 19, Table 2: Here the fonts are too large

22. Page 24, line 18 – The number 0.05 should probably be 0.5?

---

## Short Comment (SC1) · 14 Aug 2019

This is an executive editor comment on the subject of code and data availability. It highlights certain respects in which this manuscript does not currently comply with GMD model code and data policy[1]. These issues need to be remedied before a revised manuscript could be accepted for publication.

[1]https://www.geoscientific-model-development.net/about/code_and_data_policy.html

[Figure]

**Code availability**

IFS is proprietary and cannot be publicly archived. The manuscript correctly identifies this issue. However certain other code is listed as available from the author. This does not conform to GMD requirements. This code should be persistently archived, for example on Zenodo. If this is not possible for reasons beyond the control of the authors then the restrictions need to be stated (as for IFS).

**Data availability**

It is not possible to work out from the statement given which of the data on CAMS is the result of this paper. Please identify the data precisely. I presume that CAMS has a preferred mechanism for identifying and citing data sets (for example by DOI or similar), please use this mechanism if available.

---

## Referee Comment (RC2) · Anonymous Referee #2 · 16 Aug 2019

Reviewer comments to

"An aerosol climatology for global models based on the tropospheric aerosol scheme in the Integrated Forecasting System of ECMWF. Alessio Bozzo 1* , Angela Benedetti 1 , Johannes Flemming 1 , Zak Kipling 1 , and Samuel Rémy 1,2"

This is a generally well written and comprehensive paper that documents the new CAMS aerosol climatology and illustrates its application in the ECMWF forecast model. Interesting new results concerning the dynamic impact of aerosols on model results over certain areas of the globe are presented and analysed. The paper can be used as a document of the CAMS aerosol climatology data set by NWP modellers and other

users. For this, it is important to get also the details carefully presented. As not only NWP modellers are interested in the aerosol impacts, it would be good to avoid NWP-specific jargon and implicit assumptions that the reader is familiar with e.g. the data assimilation methods.

Detailed remarks and questions are presented below

p1 l6 ... set of model simulations ...

p1 l7 re-analysis or reanalysis, please check consistency throughout the paper

p1 l8 Aerosol Optical Thickness (AOT) or aerosol optical thickness, also check consistency

p1 l15 ... improve the simulation of summer monsoon circulation ... Are the words like Monsoon or Tropics or Dimethyl Sulfate written with capital letters?

p1 l24 Please check the consistency of years of both Baklanov et al. references in text/list of references

p2 l6 remove 'and' from ...prognostic aerosol field -and- because ...?

p2 l16 feed-backs of feedbacks, please check consistency

p2 l25 ... multi aerosol model simulation.. or ...multi-aerosol... ?

p2 l28 ... teleconnections ... instead of tele-connections Perhaps check all combinations of adjectives and nouns including or not including '-' ?

p4 l8-9 Dust emissions do not really depend on albedo, perhaps something like: 'in the model, emissions of dust are related to ...

p4 l10 sea salt instead of Sea-salt

p4 l13 $SO_2$ instead of SO2, mention the relation between SU and $SO_2$

p4 l16 ... an extra control variable +and+ using a variational bias correction ... ?

p4 l.19 AERONET reference, definition. You might consider an attachment table of acronyms with references?

p4 l23 ... same meteorological fields and emission +data+ as CAMSiRA ?

p4 l29 ... each specie... instead of 'species'? Or at least consistently.

p4 l31 For what you used the scaled AOT - not only for diagnostics but for something more fundamental in derivation of the mmr? Please explain in this paragraph.

p4 l32 Please explain why kg/m3 and not kg/kg as usually, e.g. in the available via CAMS near-real-time data. For this paper it may not be important as only layer-integrated values kg/m2 are shown but for data users this may be confusing.

p5 l12 What means "generally" in this sentence?

p5l 19 ... organic and black carbon species ...

p5 l23 Please discuss volcanic (stratospheric) ash and sulfates in this context: are they included in the climatology, what are the uncertainties etc. Do the dust/sulfate optical properties apply to these as well?

p5 l30 Would be logical to start from appendix A, i.e. change the order of the appendices

p5 l32 ... away from the +near-surface+ sources?

p6 Fig 1 caption ... have been multiplied by 10 ... Not the mean values shown, though?

p7 l6 ... non-negligible... ? Somewhere later you also use 'not negligible', please check consistency

p9 Fig 3 caption ... mineral dust ... ... from CR fields, the right ... ...while for organic matter +it+ is 2 km ...

p9 l5 ... emissions of black +carbon+ ...

p9 l3 Why ... it is smaller over Europe... ? Sulfates?

p9 l6 ... while showing ...

p11 l13 You have selected the sites based on dominant aerosol species. You might mention for each site what is dominating in terms of the 5 categories used here. Would an additionl Eastern European site in show in early summer something interesting related to organic (pollen etc) aerosol? Does the Karachi site show mineral (desert) dust impact? Lake Argyle seems to be in Australia, what aerosols are there? Showing a small map of the locations might also help.

p12 Table 1 CAMSiRA 2008 v.s. CAMSiRA clim remains unclear. Also further in Fig.5 you refer to CAMSiRA original. Please clarify. Is CAMSiRA (original) run for 2008 without scaling of AOT, does CAMSiRA contain your scaling?

p12 l7 Please clarify what means "compared to the IFS configuration using the old climatology based on TG97", i.e. what exactly are the differences between the configurations. See also the next comment.

p14 <l7 Please add a paragraph summarising how the radiation scheme of your experiments (Hogan and Bozzo, 2018?) treats the aerosol input in case of CAMSiRA mmr + new IOPs v.s. Tegen AOD:

- which variables enter the radiation parametrizations (AOD, SSA, ASY at each 3D gridpoint?)

- vertical distributions - native or exponential

- assumptions concerning SW and LW radiation (e.g. scattering, wavelengths really used)?

- something else?

p14 l7 What do you mean with 'model mean state' in climate runs? You only discuss the radiation fluxes, which is fine, so perhaps remove the mean state from here?
p15 l1 CERES-EBAF definition, reference (into a table of acronyms?)

p16 l17 remove extra 'on'

p17 l5 ...desert +(in China)+ ... It is perhaps Takla Makan desert?

p18 l5-10 Please reformulate this interesting list with less jargon like 'driven in part by the operator splitting of convective transport and scavenging', 'assign far too much positive increment to black carbon'

p18 l9 Please remind what are the biomass burning species

p18 l12 ... non-negligible ...?

p19 Section 4.4 is very interesting!

p19 Table 2 Definition, references to all "different products"

p20 l4 Please reformulate 'helps reducing the first-guess departure ...'

p20 l12 Is 'in the Indian Ocean' correct, or perhaps 'over'?

p21 l8 Would it be possible to say something about changes in clouds, not due to explicitly accounting for cloud-aerosol microphysics interactions but resulting anyway?

p22 l15-16 ... modifies the strength of temperature and pressure gradients over the Indian Ocean ... It seems that you did not directly show the temperature and pressure gradients but the resulting wind fields and 925 geopotential (relative topography 850-100 would directly show the mean temperature). Perhaps consider how to formulate this conclusion better.

---

## Author Comment (AC1) · 31 Oct 2019

**Answers to reviews of gmd-2019-149 "An aerosol climatology for global models based on the tropospheric aerosol scheme in the Integrated Forecasting System of ECMWF"**

**Answers to anonymous Referee 1:**

General comments:
The manuscript is well written can be useful for potential users of the CAMSiRA aerosol climatology. In particular in showing the effects of the impacts of radiation fluxes using the new climatology in the IFS forecast system the effects of the new climatology are presented.
A major concern is that the aerosol climatology is only evaluated in terms of AOT. As the new aerosol climatology was constrained by MODIS AOT, it is nice to know but unsurprising that the new CAMSiRA climatology provides a better match with AOT measurements compared to the older Tegen et al. (1997) climatology given that the older climatology was compiled from a results of very early attempts at aerosol tracer models using very coarse models and use emission fields that are meanwhile outdated. While it is a good start to look at regions that are dominated by specific aerosol types (although at most stations AOT will be a result of mixtures of different aerosol types, e.g. at Midway Island there is likely a contribution from sulfate AOT) it is notable that in particular for mineral dust evaluation at sites that are dominates by dust are absent, and should be added. As it is important for its radiative effect, particularly the effect on the Indian summer monsoon, the authors should also compare their absorbing AOT with the AERONET absorbing aerosol product. This should be a straightforward extension of the already existing analysis. Ultimately other aspects such as the mixing rations or number size distributions will be used (e.g. for simulating indirect effects of aerosol particles on clouds). Aerosol composition will play a major role for these aspects. How about comparing other aspects such as near surface concentrations?

We thank the reviewer for the thorough and very helpful comments to the manuscript.
It is true that we concentrate mostly on the evaluation of the AOT but, given that the climatology derives from the constrained CAMS reanalysis, we think that we can therefore rely on the evaluation of the CAMS model as a mean of evaluation of the climatology. Therefore we think that for a full evaluation of various other aspects of the CAMS prognostic aerosol fields, we can refer to the discussion available in Flemming et al. (2017) and Remy et al. (2019). The latter in particular also includes an analysis on particle matriculate matter (PM2.5 and PM10). We made more clear this point in the text.

In this work we are mostly interested in discussing the general impact of the climatological aerosol fields on the mean radiative fluxes in a global model, as this is what we believe would be the main use of such a database. For this we use the optical properties implemented in the ECMWF model to compute the diagnostic AOT fields, and this is only one of the possible choice available. A thorough comparison in terms of absorption optical depth would require a full study on the quality of different refractive indices for various species, which is beyond the scope of this work and the user has the freedom to specify any radiative properties of choice to associate to the CAMS aerosol species. This choice will have a significant impact especially on the most uncertain quantities such as single scattering albedo.

But we agree with the reviewer that it is indeed very useful to give an idea of how the absorption AOT in the particular implementation we discuss here, compares to the AAOT retrieved at AERONET sites. We therefore added to the general description of the spatial characteristics of the AAOD a comparison with the retrieved AAOD at selected AERONET

sites, including two new sites, one dominated mostly by the dust type and another by biomass burning in South America.

Specific comments:

1. While the introduction section gives a detailed overview about the role of aerosol climatologies in NWP and in particular for the ECMWF model, to avoid confusions the section would benefit from a table listing the current and previous aerosol climatology versions.
Table added

2. Page 4, line 32: What does 'mass volumetric concentration' mean? Do you just mean 'mass concentration'?

Corrected

3. Page 5, lines 21-25: This sentence is not clear, please explain in more detail what is meant by 'not efficient coupling' between convective transport and scavenging/ speciation/vertical distribution of analysis increments.

The paragraph has been rewritten better clarifying the concept.

4. Figure 1 The labels with numbers in some oft he panels (top 2 rows) are not explained. Are they actually needed?

The contour labels have been removed.

5. Figure 1: In addition to mass load, the distribution of the AOTs of the individual species would be interesting, as the AOTs ultimately determine the radiative effects.
This would also support the choice of Aeronet locations relevant for individual aerosol types. These locations could be indicated on such AOT maps.
Added a new figure with the AOT distribution of the individual species, including the position of the selected AERONET sites

6. Figure 4: Additional difference plots between the two climatologies would be useful to highlight their key differences.
Difference maps added to the figures

7. Page 11 and figure 5: At least one Aeronet station dominated by mineral dust should be added, as this aerosol type caused major differences between the climatologies.

Thank you, although the site of Karachi does include mineral dust, this aspect was indeed overlooked in the draft. We added one more Aeronet station affected by dust, Solar Village in Saudi Arabia which provided the most complete record for the period in question, amongst other dust-dominated sites and it is instrumental to the discussion on the indian monsoon.

8. Figure 5: what causes the dips in the green line (Tegen climatology) at the beginning of each month?

This was an artefact in the plotting script and it has been corrected.

9. Page 14, lines 10-11: Please state here for which years the 'forecast runs' are performed. In the caption of Figure 8, the period May to August of the year 2016 is named, which should also be stated in the text.

The text has been checked for consistency with the figure captions

10. Figures 6 and 7: Please provide the information on the years of the simulations in the figure captions

Information added

11. Figure 10: If, as stated in the figure caption, the figure shows also zonal winds as in Figure 9, why is the unit m2/s2 rather than m/s?

There was an error in the caption, the figure shows the geopotential, in units of m^2/s^2

Minor corrections:

12. Abstract, line 1: 'global atmospheric models' – the words should not be starting with captital letters

Corrected

13. Abstract, line 3: into -> in

Corrected

14. Abstract, line 8: : : : assimilating -the- aerosol optical thickness : : :

corrected

15. The authors use at several places in the manuscript the expression 'specie' for singular of 'species'. Please check if that is the correct usage of the singular word here. (I am not a native speaker, but would also use species for singular and plural in this context)

The reviewer is correct; we changed into species throughout the text

16. Page 6, figure 1 caption, line 1: Interim reanalysis is written as interim Reanalysis at other places in the manuscript, please make sure it is written with the same capitalization everywhere.

Corrected throughout the text

17. Page 7, line 29 ad -> and

corrected

18. Page 10, figure 4: I suggest to place the figure labels (a and b) above and not below the figures
We split the figure in two separated figures adding the panel with the differences, as per comment number 6.

19. Figure 5: The lines in figure and the labels are difficult to recognize. The lines should be thicker and the label fonts should be larger.

The figure has been improved

20. Page 17, line 10: fig -> Fig

corrected

21. Page 19, Table 2: Here the fonts are too large
fixed

22. Page 24, line 18 – The number 0.05 should probably be 0.5?

True, fixed in the text

**Answers to anonymous Referee 2:**

Reviewer comments to
"An aerosol climatology for global models based on the tropospheric aerosol scheme
in the Integrated Forecasting System of ECMWF. Alessio Bozzo 1* , Angela Benedetti
1 , Johannes Flemming 1 , Zak Kipling 1 , and Samuel Rémy 1,2"
This is a generally well written and comprehensive paper that documents the new
CAMS aerosol climatology and illustrates its application in the ECMWF forecast model.
Interesting new results concerning the dynamic impact of aerosols on model results
over certain areas of the globe are presented and analysed. The paper can be used
as a document of the CAMS aerosol climatology data set by NWP modellers and other
users. For this, it is important to get also the details carefully presented. As not only
NWP modellers are interested in the aerosol impacts, it would be good to avoid NWP
specific jargon and implicit assumptions that the reader is familiar with e.g. the data
assimilation methods.

We appreciated the reviewer detailed revision of the manuscript and the numerous
comments, which helped improving the paper. Below we provide the answers to each
specific remark.

Detailed remarks and questions are presented below

p1 l6 ... set of model simulations ...
corrected

p1 l7 re-analysis or reanalysis, please check consistency throughout the paper
consistency checked

p1 l8 Aerosol Optical Thickness (AOT) or aerosol optical thickness, also check consistency
consistency checked

p1 l15 ... improve the simulation of summer monsoon circulation ... Are the words like
Monsoon or Tropics or Dimethyl Sulfate written with capital letters?
Fixed using the correct capitalization of the words (should be lower-case)

p1 l24 Please check the consistency of years of both Baklanov et al. references in
text/list of references

corrected

p2 l6 remove 'and' from ...prognostic aerosol field -and- because ...?
removed

p2 l16 feed-backs of feedbacks, please check consistency
consistency checked throughout the text

p2 l25 ... multi aerosol model simulation.. or ...multi-aerosol... ?
consistency checked throughout the text

p2 l28 ... teleconnections ... instead of tele-connections Perhaps check all combinations
of adjectives and nouns including or not including '-' ?
consistency checked throughout the text

p4 l8-9 Dust emissions do not really depend on albedo, perhaps something like: 'in the
model, emissions of dust are related to ...
In this case the parameterization controlling the emission of dust depends on the surface
albedo to determine (together with other parameters) the points able to act as dust source
and also as a weight affecting the source strength. We clarified the text, more details are in
Remy et al. (2019).

p4 l10 sea salt instead of Sea-salt
corrected throughout the document

p4 l13 $SO_2$ instead of SO2, mention the relation between SU and $SO_2$
corrected and briefly mentioned the parametrization of the conversion rate $SO_2$->sulfate
aerosols. All the details are provided in Remy et al. (2019)

p4 l16 ... an extra control variable +and+ using a variational bias correction ... ?

The sentence here is correct, meaning that the extra control variable is implemented
adapting the bias correction framework developed for the assimilation of radiances.

p4 l.19 AERONET reference, definition. You might consider an attachment table of
acronyms with references?
Reference added. We believe that the number of acronyms is not too large as to require a
table of definitions. We checked the text for other acronyms not properly explained.

p4 l23 ... same meteorological fields and emission +data+ as CAMSiRA ?
corrected

p4 l29 ... each specie... instead of 'species'? Or at least consistently.
We checked as per other reviewer request. Species is the correct word and it has been
changed consistently throughout the document.

p4 l31 For what you used the scaled AOT - not only for diagnostics but for something
more fundamental in derivation of the mmr? Please explain in this paragraph.
The paragraph was not very clear in this respect, we agree. The scaled AOT in the context
of this work is used mainly as diagnostic. We made that clearer in the text.

p4 l32 Please explain why kg/m3 and not kg/kg as usually, e.g. in the available via
CAMS near-real-time data. For this paper it may not be important as only layer integrated
values kg/m2 are shown but for data users this may be confusing.

Indeed there was confusion here. We chose the layer-integrated mass concentration because it is directly proportional to the AOT, since we believe the likely use of such a dataset will be for radiative computations. But in the climatology we also provide gridded mean pressure profiles to allow the conversion to mass mixing ratio. We made this clear in the text.

p5 l12 What means "generally" in this sentence?
Corrected, it should have been "mostly"

p5l 19 ... organic and black carbon species ...
corrected

p5 l23 Please discuss volcanic (stratospheric) ash and sulfates in this context: are they included in the climatology, what are the uncertainties etc. Do the dust/sulfate optical properties apply to these as well?
Stratospheric aerosol of volcanic origin are not included in the climatology because not modelled in the CAMS interim reanalysis used in this work. The stratospheric residual discussed here is to a certain extent an artefact of the model, as explained. We modified the text to clarify the ambiguity

p5 l30 Would be logical to start from appendix A, i.e. change the order of the appendices
True, modified

p5 l32 ... away from the +near-surface+ sources?
corrected

p6 Fig 1 caption ... have been multiplied by 10 ... Not the mean values shown, though?
Indeed, caption clarified

p7 l6 ... non-negligible... ? Somewhere later you also use 'not negligible', please check consistency
consistency checked throughout

p9 Fig 3 caption ... mineral dust ... ... from CR fields, the right ... ...while for organic matter +it+ is 2 km ...
corrected

p9 l5 ... emissions of black +carbon+ ...

corrected

p9 l3 Why ... it is smaller over Europe... ? Sulfates?
Yes, mostly a decrease of industrial emission. Text clarified

p9 l6 ... while showing ...
corrected

p11 l13 You have selected the sites based on dominant aerosol species. You might mention for each site what is dominating in terms of the 5 categories used here. Would an additionl Eastern European site in show in early summer something interesting related to organic (pollen etc) aerosol? Does the Karachi site show mineral (desert) dust impact? Lake Argyle seems to be in Australia, what aerosols are there? Showing a small map of the locations might also help.
As per request of reviewer 1 we are now showing the position of each site over a map reporting the contribution of each species on annual mean.

p12 Table 1 CAMSiRA 2008 v.s. CAMSiRA clim remains unclear. Also further in Fig.5 you refer to CAMSiRA original. Please clarify. Is CAMSiRA (original) run for 2008 without scaling of AOT, does CAMSiRA contain your scaling?
We modified the text, hopefully clearer now

p12 l7 Please clarify what means "compared to the IFS configuration using the old climatology based on TG97", i.e. what exactly are the differences between the configurations.
See also the next comment.
p14 <l7 Please add a paragraph summarising how the radiation scheme of your experiments (Hogan and Bozzo, 2018?) treats the aerosol input in case of CAMSiRA mmr + new IOPs v.s. Tegen AOD:
- which variables enter the radiation parametrizations (AOD, SSA, ASY at each 3D gridpoint?)
- vertical distributions - native or exponential
- assumptions concerning SW and LW radiation (e.g. scattering, wavelengths really used)?
- something else?

An extra paragraph was added better explaining the two configurations

p14 l7 What do you mean with 'model mean state' in climate runs? You only discuss the radiation fluxes, which is fine, so perhaps remove the mean state from here?

'model mean state' was indeed out of context and it has been removed

p15 l1 CERES-EBAF definition, reference (into a table of acronyms?)
acronym explained and added relevant references

p16 l17 remove extra 'on'
fixed

p17 l5 ...desert +(in China)+ ... It is perhaps Takla Makan desert?
We found it spelled in various ways; Taklamakan seems to be the one used more often

p18 l5-10 Please reformulate this interesting list with less jargon like 'driven in part by the operator splitting of convective transport and scavenging', 'assign far too much positive increment to black carbon'
We clarified the paragraph

p18 l9 Please remind what are the biomass burning species
there was some confusion throughout the text between organic species and biomass burning, which is part of the organic species. We clarified the composition of the organic matter species and corrected the text

p18 l12 ... non-negligible ...?
corrected

p19 Section 4.4 is very interesting!
Thank you

p19 Table 2 Definition, references to all "different products"
references and full acronyms explanations added to the table

p20 l4 Please reformulate 'helps reducing the first-guess departure ...'
modified

p20 l12 Is 'in the Indian Ocean' correct, or perhaps 'over'?
corrected

p21 l8 Would it be possible to say something about changes in clouds, not due to explicitly accounting for cloud-aerosol microphysics interactions but resulting anyway?
We added a short paragraph in this section linking to the results observed in section 4.2

p22 l15-16 ... modifies the strength of temperature and pressure gradients over the Indian Ocean ... It seems that you did not directly show the temperature and pressure gradients but the resulting wind fields and 925 geopotential (relative topography 850-100 would directly show the mean temperature). Perhaps consider how to formulate this conclusion better.

We modified the conclusions so they reflect better what we showed in the previous section. Although the impact on the monsoon circulation is very interesting, it was shown here just as an example of potential impacts that can be expected when modifying the aerosol radiative effect and unfortunately a deeper discussion on the topic is beyond the scope of this technical paper.

**Answers to executive editor comment:**

This is an executive editor comment on the subject of code and data availability. It highlights certain respects in which this manuscript does not currently comply with GMD model code and data policy. These issues need to be remedied before a revised manuscript could be accepted for publication.

**Code availability**
IFS is proprietary and cannot be publicly archived. The manuscript correctly identifies this issue. However certain other code is listed as available from the author. This does not conform to GMD requirements. This code should be persistently archived, for example on Zenodo. If this is not possible for reasons beyond the control of the authors then the restrictions need to be stated (as for IFS).
Thank you for the comment. The code mentioned is actually a pretty standard algorithm to compute the scattering properties of spheres with a defined refractive index and we realised there is no need to release this particular code publicly, also given the complications due to the restrictions that computer codes are subjected to when developed at ECMWF. We listed the appropriate reference for the algorithm.

**Data availability**
It is not possible to work out from the statement given which of the data on CAMS is the result of this paper. Please identify the data precisely. I presume that CAMS has a preferred mechanism for identifying and citing data sets (for example by DOI or similar), please use this mechanism if available.

Indeed the section was not clear. We clarified which datasets will be available and how to access it. Data on the CAMS archive do not have a DOI associated so we indicated the location where the data will be stored and a point of contact. We would like upload the data and provide the complete address once the revision process is completed.

---

## Author Response (AR2)

**Answer to Topical Editor comment about data availability**

Dear Topical Editor,

Thank you for your comments clarifying the requirements for the data availability. Please find below the answers addressing the specific points in your comments followed by the manuscript with the track changes highlighting the correction in the data availability section.

*1. Your response that the two datasets as a consequence of this work "will" be uploaded once the review process is complete, unfortunately, does not meet the requirements of the GMD data policy. May I please ask you to upload the relevant datasets to the repository?*

*2. Secondly, may I please ask that you provide me with a very precise location for each of the data sets on the repository and revise the data section of your paper accordingly?*

Data have been uploaded to the repository with an introduction page describing the datasets and the relative references. The page can be found at this address:

https://confluence.ecmwf.int/display/CKB/CAMS+Monthly+Mean+Aerosol+Climatology+for+global+models

and the two datasets at the respective locations:

climatology file: ftp://ftp.ecmwf.int/pub/cams/datasets/aerosol_radiation_climatology/2003-2013/aerosol_cams_3d_climatology_2003-2013.nc

radiative optics file: ftp://ftp.ecmwf.int/pub/cams/datasets/aerosol_radiation_climatology/aerosol_cams_ifs_optics.nc

We revised the data section accordingly.

Kind Regards,

Alessio

[revised manuscript text omitted]

---

## Author Response (AR3)

**Answer to Topical Editor comment about browser-independent data availability**

Dear Topical Editor,

To address the issue of browser-dependent ftp link behaviour, we modified the location where data are stored. The data availability section was modified accordingly.

The data are now available for download at the permanent address https://sites.ecmwf.int/data/cams/aerosol_radiation_climatology/ with the associated DOI https://doi.org/10.24380/jgs8-sc58.

and the link to the two datasets are:

climatology file

https://sites.ecmwf.int/data/cams/aerosol_radiation_climatology/2003-2013/aerosol_cams_3d_climatology_2003-2013.nc

radiative optics file

https://sites.ecmwf.int/data/cams/aerosol_radiation_climatology/aerosol_cams_ifs_optics.nc

Kind Regards,

Alessio

[revised manuscript text omitted]